# Concurrence of form and function in developing networks and its role in synaptic pruning

Ana P. Millán[1], J.J. Torres[1], S. Johnson [2] & J Marro [1]

A fundamental question in neuroscience is how structure and function of neural systems are related. We study this interplay by combining a familiar auto-associative neural network with an evolving mechanism for the birth and death of synapses. A feedback loop then arises leading to two qualitatively different types of behaviour. In one, the network structure becomes heterogeneous and dissasortative, and the system displays good memory performance; furthermore, the structure is optimised for the particular memory patterns stored during the process. In the other, the structure remains homogeneous and incapable of pattern retrieval. These findings provide an inspiring picture of brain structure and dynamics that is compatible with experimental results on early brain development, and may help to explain synaptic pruning. Other evolving networks—such as those of protein interactions—might share the basic ingredients for this feedback loop and other questions, and indeed many of their structural features are as predicted by our model.

[1] Institute Carlos I for Theoretical and Computational Physics, University of Granada, Granada 18010, Spain. [2] School of Mathematics, University of Birmingham, Edgbaston B15 2TT, UK. Correspondence and requests for materials should be addressed to A.P.Mán. (email: apmillan@ugr.es)

The fundamental question in neuroscience of how structural and functional properties of neural networks are related has recently been considered in terms of large-scale connectomes and functional networks obtained with various imaging techniques[1]. But at the lower level of individual neurons and synapses there is still much to learn. The brain can be regarded as a complex network in which nodes represent neurons, and edges stand for synapses. It is then possible to use mathematical models, which capture the essence of neural and synaptic activity, to study how a great many of such elements can give rise to collective behaviour with at least some of the characteristics of cognition[2–4]. In this context, the structural properties of the underlying network supporting brain dynamics have been found to affect the behaviour of the system in various ways[5–8]. For instance, experimental setups have confirmed that actual neural networks exhibit degree heterogeneity that roughly accords with scale-free distributions, and negative degree–degree correlations (dissasortativity), which strongly influence the dynamics of the system[9,10].

These networks are also not static: new synapses grow and others disappear in response to neural activity[11–14]. In many species, early brain development seems to be dominated by a remarkable process known as synaptic pruning. The brain starts out with a relatively high density of synapses, which is gradually reduced as the individual matures. In humans, for example, synaptic density at birth is about twice what it will be at puberty, and certain disorders, such as autism and schizophrenia, have been related to details of this process[15–19]. It has been suggested that such synaptic pruning may represent some kind of optimization, perhaps minimizing energy consumption and/or the genetic information needed to build an efficient and robust network, and/or to optimize network structure[20,21].

Some progress has been made in understanding how new synapses are grown. It has been shown that pre-synaptic activity and glutamate could act as trophic factors to guide new synaptic spines growth[22], and other mechanisms of cooperation among neurons have also been proposed, such as spike-timing dependence plasticity[23,24]. Imaging experiments also reveal that brain architecture is sculpted by spontaneous and sensory-evoked activity, in a process that goes on into adulthood, as synaptic circuits continue to stabilize in the mature brain[22,25]. Even though synapses are highly dynamic in time, their overall statistics are preserved over time in the adult brain, indicating that synapse creation and pruning balance each other[23].

In this context, models in which networks are gradually formed, for instance by addition of nodes and edges or by rewiring of the latter, have long been studied in different contexts. Typically, the probabilistic addition or deletion of elements is a function of the existing structure. For example, in the familiar Barabási–Albert model, a node's probability of receiving a new edge is proportional to its degree[26]. These rules often give rise to phase transitions (almost invariably of a continuous nature), such that different kinds of network topology can ensue, depending on parameters[27], and have been used in the past to reproduce some connectivity data on human brain development[28]. More recently, models in which the evolution of network structure is intrinsically coupled with an activity model that runs on the nodes of the network—the so-called co-evolving network models—have gained attention as a way to approximate the evolution of real systems[29–31]. Previous studies of co-evolving brain networks have studied the temporal evolution of mean degree[32], particular microscopic mechanisms[20], the development of certain computational capabilities[33], or the effects of specific growth rules[34], and have suggested evidence for the role of bistability and discontinuous transitions in the brain, for instance in synaptic plasticity mechanisms involved in learning[35,36].

Here we define a co-evolving model for brain network development by combining the Amari-Hopfield neural network[2,3] with a plausible model of network evolution[28], by setting the probabilities of synaptic growth and death to depend on neural activity, as it has been empirically observed[22]. We find that, for certain parameter ranges, the phase transition between memory and randomness becomes discontinuous (i.e., resembling a first order thermodynamic transition). Depending on initial conditions, the system can either evolve towards heterogeneous networks with good memory performance, or homogeneous ones incapable of memory, as a consequence of the feedback loop between structure and function. To the best of our knowledge, this is the first time that this feedback loop, and the ensuing discontinuous transition, have been identified. Also, in our model networks are generated, which have optimal memory performance for the specific memory patterns they encode, thus allowing for a greater memory capacity than would be possible in the absence of such a mechanism. Our results thus suggest a more complete explanation of synaptic pruning. Finally, we discuss the possibility that other biological systems—in particular, protein interaction networks—also owe their topologies to a version of the feedback loop between form and function that we identify here.

## Results

**Model construction**. We define a co-evolving model of synaptic pruning that couples a traditional associative memory model, the Hopfield model[37], with a preferential attachment model for network evolution[28]. Memory is measured via the overlap, $m^\mu(t)$, of the network with the $P$ memorized patterns of activity, $\{\xi^\mu\}$, which are stored by an appropriate definition of the synaptic weights $w_{ij}$. In a stationary regime, it undergoes a continuous transition from a phase of memory recovery ($m^\mu \to 1$) to one dominated by noise ($m^\mu \approx 0$) as a function of the noise level or temperature $T$.

Network structure dynamics is defined by the probability each node $i$ has to gain or to lose an edge:

$$P_i^g = u(\kappa)\pi(I_i), \quad P_i^l = d(\kappa)\eta(I_i), \quad (1)$$

respectively, where $\kappa$ is the mean degree of the network and $I_i$ a physiological variable that measures the incoming current at node $i$. A second node $j$ is then randomly chosen to be connected to (or disconnected from) node $i$, so that there are two processes that can lead to an increase (decrease) in node degree, and we shall define the effective local probabilities $\tilde\pi$ and $\tilde\eta$ to account for both of them (see the Methods). We choose these to be power-law distributed, $\tilde\pi \propto I_i^\alpha$ and $\tilde\eta \propto I_i^\gamma$, to allow for a smooth transition from a sub-linear to a super-linear dependence with a single parameter. Network structure is characterized by the homogeneity parameter $g(t)$, which equals 1 if $p(k) = \delta_{k_0, k}$ and tends to 0 for highly heterogeneous (bimodal) networks. Depending on $\alpha$ and $\gamma$, networks are homogeneous (every node having similar degree, $g \to 1$) or heterogeneous (with the appearance of hubs, $g \to 0$). Probabilities $u$ and $d$ are chosen to be consistent with empirical data on synaptic pruning, as we discuss in subsection Synaptic pruning. A full and comprehensive description of the model is detailed in the Methods.

**Topological limit**. Consider first the topological limit of the model as defined by Eq. (4), in which network structure is decoupled from neural activity. Given the biologically inspired probabilities of Eq. (2) and (3), three qualitatively different behaviours, leading in practice to different phases, are possible for $k_i \gg 1$ (Fig. 1a). If $\gamma > \alpha$, then $\tilde\eta(k_i) > \tilde\pi(k_i)$, and high-degree nodes are more likely to lose than to gain edges, so that

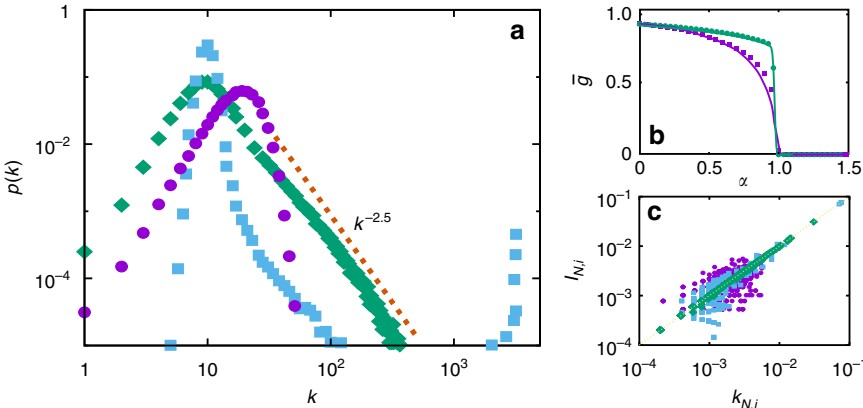

**Fig. 1** Topological limit and coupling. **a** Stationary degree distributions $p_\infty(k)$ for the three different phases: subcritical ($\alpha = 0.8$, purple circles), critical ($\alpha = 1.0$, green diamonds) and supercritical ($\alpha = 1.2$, blue squares). **b** Stationary homogeneity $\bar{g}(\alpha)$ for the topological limit ($\gamma = 1.0$, purple squares), and for a different set of local probabilities ($\pi(k) = k^\alpha/(\langle k^\alpha \rangle N)$ and $\eta(k)$ as before, green circles). The transition in the latter case is a discontinuous one, which fails to produce scale-free networks. The indicated lines follow from integration of the corresponding master equations, whereas points are Monte Carlo data. Deviations from simulations are due to the emergence of small degree–degree correlations that are not taken into account by the master equation. **c** Normalized current of each neuron, $I_{N,i}$, as a function of its normalized degree, $k_{N,i}$, showing the correlation between neural activity and topology that emerges in the system. The normalization is made by averaging over the total current ($\langle I \rangle N$) and degree ($\langle k \rangle N$) of the network, respectively. Labels as in **a**. Results are for $N = 3200$ and $\gamma = 1.0$, and data from Monte Carlo simulations have been averaged over 100 realizations. Error bars corresponding to s.d. are too small to be appreciable

$p_\infty(k) = p(k,t\rightarrow\infty)$ is homogeneous, $g(t\rightarrow\infty)\rightarrow1$, and the probability of having high-degree nodes vanishes rapidly from a maximum. On the other hand, if $\gamma<\alpha$, then $\tilde{\eta}(k_i)<\tilde{\pi}(k_i)$, and high-degree nodes are more likely to continue to gain than to lose edges. Since the stationary number of edges $N\kappa_\infty$ is fixed, this leads to a bimodal $p_\infty(k)$, with $g(t\rightarrow\infty)\rightarrow0$. Finally, in the case $\gamma = \alpha$, $\tilde{\eta}(k_i) = \tilde{\pi}(k_i)$ and very connected nodes are as likely to gain as to lose edges. Excluding low degrees, $p_\infty(k)$ then decays as a power-law with exponent $\mu \approx 2.5$, in accordance with long-range connections observed in the human brain[8] and measures in protein interaction networks[38]. We have found that this condition is mandatory to obtain critical behaviour, despite previous preliminary studies assuming the contrary[28], as shown in Fig. 1b.

**Synaptic pruning**. The time evolution of $\kappa(t)$ is controlled in our model by the global probabilities $u(\kappa)$ and $d(\kappa)$, which can be chosen to model experimental data on synaptic density during brain development[28]. In particular, here we analyze and fit two experimental data sets (Fig. 2): the first one (a) corresponds to postmortem measures on layers 1 and 2 of the human auditory cortex, obtained by directly counting synapses in the tissue[32]; whereas, the second set (b) corresponds to an electron microscopy imaging study on the mouse somatosensory cortex[21]. Even though they correspond to different animals and have been obtained through different techniques, both data sets show the same overall behaviour: an extreme initial growth of synapses, followed by a maximum when pruning begins and connectivity starts decreasing, until a plateau is reached. Synaptic density decays roughly exponentially during pruning, and can be fitted by $u(\kappa)$ and $d(\kappa)$ as given by Eq. (2), which describe a situation in which synapses are less likely to grow, and more likely to atrophy, when the connectivity is high, and vice versa, a situation that could easily arise in the presence of a finite quantity of nutrients. These lead to $\kappa(t) = (\kappa_0-\kappa_\infty)\exp(-t/\tau_p)+\kappa_\infty$, where $\tau_p = N\kappa_\infty/(2n)$, so that $\kappa(t)$ decays exponentially from $\kappa_0$ to $\kappa_\infty$, assuming that $\kappa_0>\kappa_\infty$ as in the case of interest.

On the other hand, the initial overgrowth of synapses can be related to the transient existence of some growth factors, and it can be accounted for in our model by including a nonlinear, time dependent term, $c(t) = a \exp(-t/\tau_g)$, in the growth probability $u$

($\kappa$). The solution is now $\kappa(t) = \kappa_\infty[1+b \exp(-t/\tau_g)-a \exp(-t/\tau_p)]$, with $b = a\tau_g/(\tau_g-\tau_p)$, $a = 1-\kappa_0/\kappa_\infty+b$ and $\tau_p$ as before. With the inclusion of this term, the evolution of $\kappa(t)$ on both data sets can be fully reproduced.

Therefore, our model can approximate the evolution of the mean density of synapses in the mammal brain during infancy, and with the inclusion of an initial growth factor it also reproduces the fast growth and early maximum of the connectivity. Notice that this framework also goes in line with previous studies that have highlighted the computational benefits of a pruning process in which the pruning rate decreases with time, as in our model, which can optimize both efficiency and robustness when growth takes place locally throughout the network[21]. In our model the decreasing rate is naturally obtained via a simple, physiologically inspired master equation for $p(k,t)$. Even when an extra growth factor is included, once pruning begins its rate decreases over time, in accordance with[21]. Given that our goal here is to understand the effect of neuronal physiological activity on network development, in the framework of co-evolving growing models, we shall focus on the simplest version of the model, with linear dependence on $u(\kappa)$ and $d(\kappa)$, to illustrate the effect of coupling structure and physiology. Nonlinear global probabilities could also be considered, but would add an extra level of complexity which we will not explore here.

**Phase diagram**. In the coupled model, where pruning depends on $I_i$ (see the Methods section), computer simulations depict a rich emergent phenomenology depending on stochasticity ($T$) and emerging degree heterogeneity ($\alpha$). The effect of the number of memorized patterns $P$ is analyzed in subsection Capacity analysis, and we discuss the effect of $\gamma$ in Supplementary Fig. 1. Other parameters, such as $\kappa_\infty$ or $a_0$, were found to have only a quantitative effect on the resulting phase diagram, and they are set as in a previous study[28]. Preliminary simulations suggested dependence on the heterogeneity of the initial condition (IC), so we consider two different types of IC, namely, homogeneous, $p(k,t=0) = \delta(k-\kappa_0)$, and heterogeneous, $p(k,t=0)\propto k^{-2.5}$, networks, with fixed $\kappa_0$.

Our coupling dynamics lead to a rich phenomenology, including discontinuous transitions and multistability. Three

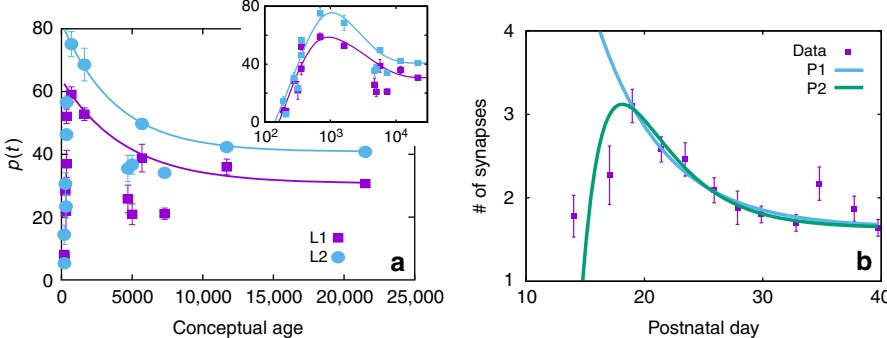

**Fig. 2** Synaptic pruning. Experimental data sets on connectivity during infancy (points) and model fit (solid lines). **a** Data points correspond to synaptic density, $\rho(t) \propto \kappa(t)$, in the human infant brain, obtained in autopsies by directly counting synapses in tissues from different layers of the auditory cortex (here shown layers 1 (L1) and 2 (L2))[32]. The solid lines present the best fit obtained by the model, with $\tau_{p,L1} = 1600(300)$ and $\tau_{p,L2} = 3800(100)$. Other parameters are extracted from the data: L1: $\kappa_0 = 59.1$, $\kappa_\infty = 30.7$ and $t_0 = 700$; and L2: $\kappa_0 = 75.1$, $\kappa_\infty = 40.8$ and $t_0 = 700$, where $t_0$ is set at the onset of pruning (i.e., corresponding to the maximum of $\kappa(t)$). Inset on the right shows the fit of the maximum on a log–log scale, labels as in the main plot. Parameters from the fit are $a_{L1} = 33(5)$, $\tau_{g,L1} = 210(20)$, $\tau_{p,L1} = 3800(200)$, $a_{L2} = 1.2(2)$, $\tau_{g,L2} = 2700(100)$, $\tau_{p,L2} = 290(40)$. The three lower points for $t \approx 5000$ have been excluded from the fits. **b** Data points correspond to synaptic density measured via large-scale brain imaging experiments that quantify the number of connections in the developing mouse somatosensory cortex[21]. In blue solid lines (P1), fit with the linear model, for parameters $\tau_p = 5.72(1)$, $\kappa_0 = 3.10$, $\kappa_\infty = 1.64$ and $t_0 = 18.97$. In green solid lines (P2), fit including the growth factor, with parameters $a = 1.7(1)$, $\tau_g = 4.0(2)$ and $\tau_p = 2.2(2)$. The third point form the right has been excluded from the fits. Error bars of the data points correspond to s.d. as obtained in the original works

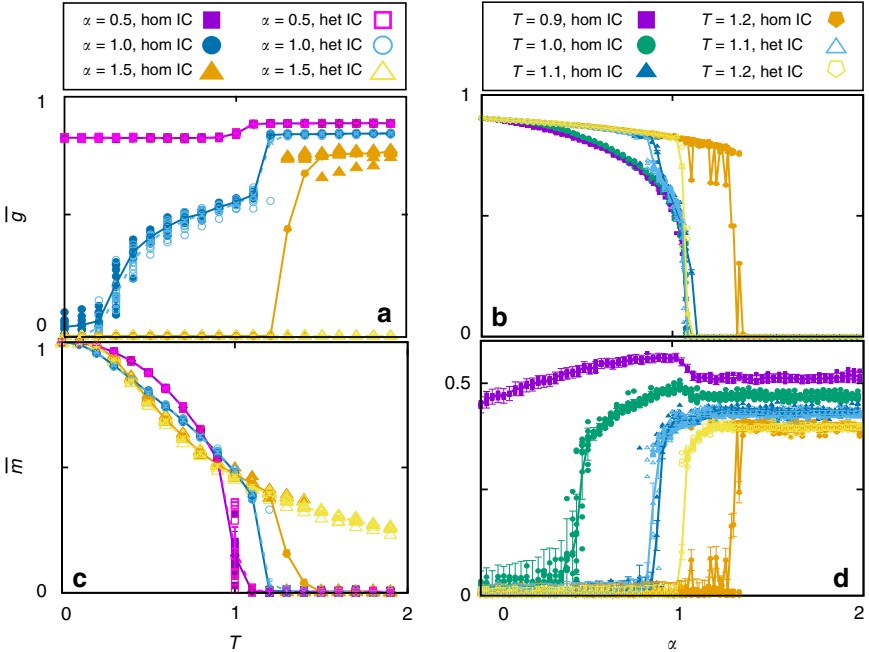

**Fig. 3** Transition lines. Behaviour along the dotted lines marked in Fig. 4, for $N = 1600$, $\kappa_\infty = 10$ and $n = 5$. **a** and **c** show $\overline{g}(T)$ and $\overline{m}(T)$, respectively, for different values of $\alpha$, as indicated. **b** and **d** correspond, respectively, to $\overline{g}(\alpha)$ and $\overline{m}(\alpha)$, for different temperatures as indicated. Solid points are for homogeneous IC and empty ones for heterogeneous IC, in every panel. Notice that some isolines go through the three phases: for $T = 1.0$ and 1.1 (**b** and **d**) the system is initially in the homogeneous memory phase, and an increase in $\alpha$ leads through $\alpha_c^m(T)$ to networks that are heterogeneous enough to maintain memory, which further increases heterogeneity due to the feedback loop ($\overline{g}(\alpha)$ decreases). A final increase in $\alpha$ leads through $\alpha_c^t(T)$ to heterogeneous networks. Similarly in **a** and **c**, the system visits for $\alpha = 1$ the three phases: at very low $T$, $\alpha$ is high enough to develop heterogeneous memory networks, but a slight noise increase is enough to suppress heterogeneity, until noise is too high and memory is also lost, leading to the homogeneous noisy phase. Data points are averaged over 30 realizations and error bars correspond to s.d

different phases can be identified by monitoring the stationary values of the order parameters, $\overline{g}(\alpha, T)$, $\overline{m}(\alpha, T)$, and the Pearson correlation coefficient, $\overline{r}(\alpha, T)$, as illustrated in Fig. 3 (data for $\overline{r}(\alpha, T)$ is not shown here since it provides similar information as $\overline{g}(\alpha, T)$). Analysis of these and similar curves for different parameter values leads to the phase diagram in Fig. 4. That is, there is a homogeneous memory phase for low $\alpha$ and $T$ which is characterized by high $\overline{m}$, high $\overline{g}$ and low (negative, almost zero) $\overline{r}$

(Fig. 3a, c for $\alpha = 0.5$ and low $T$, and in Fig. 3b, d for $T < 1.2$ and low $\alpha$). The system is then able to reach and maintain memory, while the topological processes lead to a homogeneous network configuration. Due to the existence of memory, there is a strong correlation between the physiological state of the network, as measured by the currents $I_i$, and its topology, as reflected by the degrees $k_i$ (Fig. 1c).

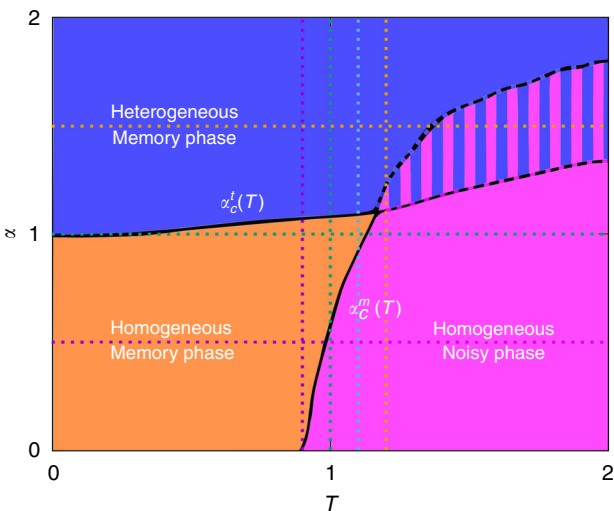

**Fig. 4** Phase diagram. Phase diagram as obtained from analysis of the control parameters ($\overline{m}$, $\overline{g}$ and $\overline{r}$) as defined in the main text. The transition lines, corresponding to $\alpha_c^t(T)$ and $\alpha_c^m(T)$ as labelled, are indicated either with solid or dashed lines regarding whether they correspond to continuous or discontinuous transitions. The horizontal and vertical dotted lines correspond to the cases illustrated in Fig. 3. Parameters as in Fig. 3. Data points are averaged over 10 realizations

A continuous increase in $\alpha$ leads through a topological phase transition from homogeneous final networks (with roughly Poisson degree distributions) to heterogeneous ones (with bimodal degree distributions). At the critical point $\alpha = \alpha_c^t(T)$ the emergent networks are scale-free (i.e., with power-law-like degree distributions). The phase transition is revealed in $\overline{g}(\alpha)$ (Fig. 3b for $T = 0.9$, 1.0 and 1.1) as a fast (and continuous) decay to zero, and it also appears in $\overline{m}(\alpha)$ (Fig. 3d) where, after an initial growth with $\alpha$, $\overline{m}$ then decreases at the transition, and finally approaches a constant value for $\alpha > \alpha_c^t(T)$. This is a consequence of the strong coupling between structure and memory, and it shows that scale-free networks optimize memory recovery for a given set of control parameters. We call this the heterogeneous memory phase, characterized by high $\overline{m}$, very low (almost zero) $\overline{g}$ and high negative $\overline{r}$, indicating a memory state with heterogeneous (bimodal) dissassortative structure. Interestingly, this phase expands up to high-noise levels as a consequence of network heterogeneity, which increases memory performance[9,10]. Moreover, memory recovery in turn favours network heterogenization in a feedback manner due to the microscopic dynamics, enhancing the stability of the state. This is because $I_i$ becomes proportional to $k_i$ in the memory regime, whereas this correlation is reduced in disordered neural states (see Fig. 1c).

As $T$ is further increased, dynamics is finally governed by noise, and the stationary network is then homogeneous, resulting in the homogeneous noisy phase, characterized by low (almost zero) $\overline{m}$, high $\overline{g}$ and low (almost zero) $\overline{r}$ (Fig. 3a, c for $\alpha = 0.5$ and 1.0 and high $T$, and Fig. 3b, d for $T > 0.9$ and low $\alpha$).

A particularly interesting aspect of this phenomenology is that the nature of the phase transition with $T$ depends on $\alpha$. For low $\alpha$ ($\alpha < \alpha_c^t(T)$) there is a continuous (second order) transition with increasing $T$ from a homogeneous memory phase to a homogeneous noisy one through $\alpha = \alpha_c^m(T)$ (Fig. 3a, c for $\alpha = 0.5$). On the other hand, at higher $\alpha$ ($\alpha > \alpha_c^t(T)$) the transition from the heterogeneous memory phase to the homogeneous noisy one is discontinuous (first order), and includes a bistability region (striped area in Fig. 4) in which simulations starting from heterogeneous IC reach the heterogeneous memory state, whereas

those starting from homogeneous ones fall into the homogeneous noisy one (Fig. 3b, d for $T = 1.2$, and Fig. 3a, c for $\alpha = 1.5$).

The existence of multistability illustrates how memory promotes itself in a heterogeneous network, which is a direct consequence of the coupled dynamics in our model, and it is lost in the topological limit (see Supplementary Fig. 1). This is because heterogeneous networks present higher memory recovery than homogeneous ones for high-noise levels, particularly for $T > 1$[9,10]. At the same time, given that in our model the growth and death of links depend on the activity of the nodes, the evolution of the structure of the network is driven by its activity state. In this way, an ordered state of the activity of the system—that is, a memory state—is needed in order to allow for the formation of heterogeneous (ordered) structures. Hence, if the network is in a noisy state, edge birth and death are random processes, and thus lead to a homogeneous network configuration, whereas in a memory state there is a direct correlation between $I_i$ and $k_i$ (Fig. 1b) that allows for the emergence of structure—as given by the local probabilities. Correspondingly, the physiological state directly depends on the network structure through the currents $I_i$, thus closing a memory-heterogeneity feedback loop. As a consequence, homogeneous and heterogeneous IC evolve differently, which translates into a multistability region for high $\alpha$ and $T$, and the presence of memory for $T > 1$.

Therefore, a main observation here is that the model shows an intriguing relationship between memory and topology, which induces complex transitions that might be relevant for understanding actual systems. In particular, the inclusion of a topological process allows for memory recovery when $T > 1$; whereas, the presence of thermal noise shifts the topological transition, which in the topological limit occurs at $\alpha = 1$, to $\alpha > 1$. These findings can also have implications for network design—for instance, to help memory recovery optimization in noisy environments. It is worth noting also that one does not need to know the specific patterns that induce the synaptic weights in order to have a memory state, so that the definition of memories is not essential—only an ordered state of neural activity. Therefore, our results could be extended to other models of microscopic activity, not necessarily based on a Hebbian learning, or to other systems such as protein interaction networks (see subsection Protein interaction networks), as long as they present a transition between an ordered and a disordered activity state.

**Capacity analysis.** We have so far considered $P = 1$ for the sake of simplicity. However, for neural networks—whether natural or artificial—to be useful, they must generally store many different memory patterns. Something analogous seems also to be true for other complex systems, such as gene regulatory networks, which switch between different configurations. We therefore go on to study the capacity of the network—that is, how its memory capability depends on the number of memorized patterns $P$. For random uncorrelated memory patterns ($a_0 = 0.5$) the performance (as measured by the overlap $\overline{m}$ of the recovered pattern) drops fast with $P$[39] (see Fig. 5a). However, there is experimental evidence that the configurations of neural activity related to particular memories in the animal brain involve many more silent neurons, $\xi_i^\mu = 0$, than active ones, $\xi_i^\mu = 1$[40] (notice that in this case there will be a positive correlation between different patterns); we therefore consider this kind of activity patterns in Fig. 5b, and find that good memory performance can be maintained at higher $P$ depending on $\alpha$. These curves are combined with analogous ones for $\overline{g}(P, \alpha)$ to define the phase diagram of panel c. This shows that for $\alpha < 1$ memory is preserved only for small $P$ and the structural noise (or quenched disorder) introduced by the patterns has a similar effect to that of the thermal

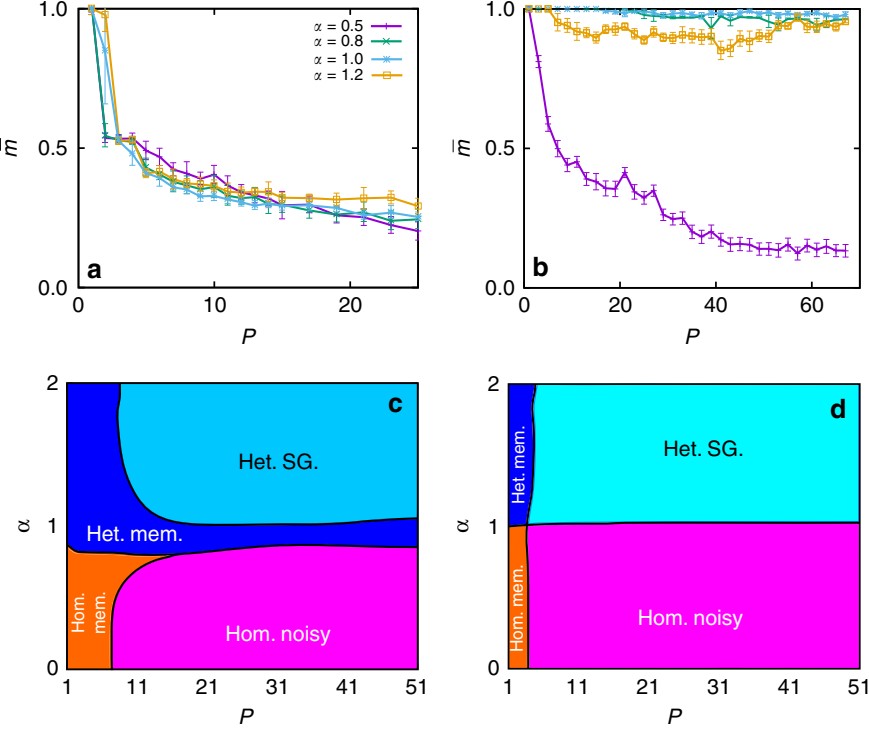

**Fig. 5** Capacity analysis. Capacity curves ($T = 0$) for (**a**) randomly distributed patterns ($a_0 = 0.5$), and (**b**) patterns distributed in small sets of active neurons ($a_0 \approx 1/P$). $\overline{m}$ corresponds to the recovered pattern in each case. Results are for $N = 800$ and are averaged over 50 realizations, with other parameters as before. Error bars correspond to the s.d. **c** and **d** show the phase diagram of the system, obtained as in Fig. 4, as a function of $\alpha$ and $P$ for the full model (**c**) and the topological limit (**d**). Data points have been averaged over 10 realizations

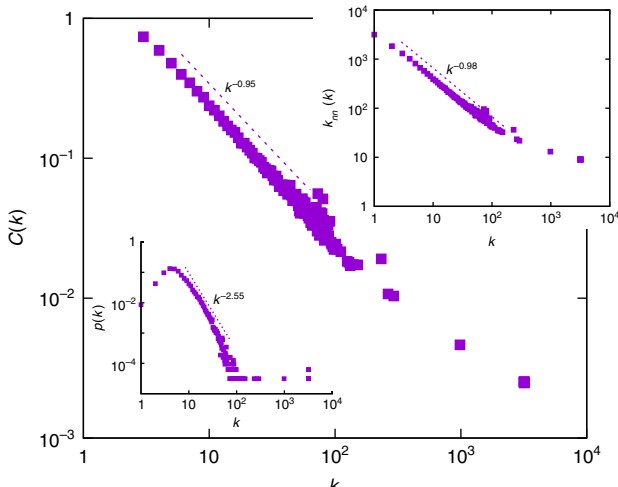

**Fig. 6** Protein interaction networks. Statistical comparison with protein interaction networks: clustering coefficient $C(k)$ (main plot), neighbours mean degree $k_{nn}(k)$ (top inset) and probability distribution $p(k)$ (bottom inset) along the critical transition line $\alpha_c^t(T)$ ($T = 0.5$, $\alpha = 1.05$). We include a power-law fit of the tails, the exponents are: $\mu_M = 2.55(1)$, $\nu_M = 0.98(1)$ and $\theta_M = -0.95(2)$, respectively, for $p(k)$, $C(k)$ and $k_{nn}(k)$. Data points are averaged over 100 realizations

noise in Fig. 4, and a continuous (second order) transition from homogeneous memory to homogeneous noisy networks is observed. On the other hand, for $\alpha > 1$ the competition between different patterns boosts network heterogeneity, and the capacity of the network increases greatly. There is a transition from a pure memory state to a spin-glass-like state (SG), in which several patterns are partially retrieved at the same time. However, due to the presence of hubs and the correlation among patterns, this corresponds to high overlap ($\overline{m^\mu} \approx 1$ for the recovered patterns), instead of a moderate value as one would expect in a typical spin-glass phase[37].

Finally, Fig. 5d shows the same diagram but for the topological limit of the model, where now the memory phases are much more narrow and SG states correspond to moderate overlap values ($\overline{m^\mu} \approx 0.6$), thus indicating a pure SG state. Therefore, the capacity analysis reveals another significant difference between the topological and coupled versions of the model, since the coupled one allows for good memory retrieval even at much larger numbers of patterns ($P > 50$) for $\alpha \simeq 1$—that is, for the limiting case in which the emerging networks are approximately scale-free. It follows that the emerging network topologies in the coupled version do not owe their good memory performance solely to their power-law degree distributions. Rather, the network is tuned for good memory performance on the specific patterns encoded in the edge weights.

**Protein interaction networks**. Many complex systems can be described as networks which evolve under the influence of node activity, and it is likely that the described structure-function feedback loop plays a role in these settings too. In particular, it could be compatible with experimental and theoretical studies concerning protein interaction networks, which gather different types of metabolic interactions amongst proteins. These can be either inhibitory or excitatory and also have different strengths, as the edges in our model. Moreover, network structure changes on an evolutionary timescale, during the evolution of the species, and these changes combine a random origin, typically due to mutations, with a "force" driven by natural selection, which is likely to be activity dependent[38]. The similarities between this picture and

our model suggest a parallelism concerning the resulting network topologies as well.

Measurements on protein interaction networks show power-law distributions of some important topological magnitudes: degree distribution $p(k) \propto k^{-\mu}$, with $\mu \approx 2.5$[38]; clustering coefficient $C(k) \propto k^{-\theta}$, with $\theta \approx 1$ (metabolic networks) or 2 (protein interaction networks)[38,41]; and neighbours mean degree $k_{nn}(k) \propto k^{-\nu}$, with $\nu \approx 0.6$[42]. We find that these magnitudes are also power-law distributed for networks in our model near the transition $\alpha_c^t(T)$ (Fig. 6), with exponents $\mu_M = 2.55 \pm 0.01$, $\theta_M = 0.98 \pm 0.01$ and $\nu_M = 0.95 \pm 0.02$, so there is a good agreement with $\mu$ and $\theta$. Moreover, $k_{nn}(k)$ decays in our model as a power-law for almost every value of the parameters, which is related to the intrinsic topological dynamics of the model, that creates an asymmetry between the nodes that gain and loose edges[42]. We find $\nu_M \approx 0.5$ for homogeneous networks and $\nu_M \approx 1.5$ for bimodal ones, whereas scale-free networks lie in between, with $\nu_M \approx 1.0$. It is likely that this parameter could be better reproduced with further adjustments in the local probabilities, so as to reflect degree–degree correlations among different proteins.

In conclusion, there are definite indications that some of the main topological properties of protein interaction networks could be qualitatively reproduced with simple adjustments or extensions of our model. This suggests a general mechanism underlying the dynamics of different biological systems, which is likely to extend as well to other contexts. Moreover, several studies have recently shown that there are specific patterns in protein interaction networks that can be determined experimentally[50,51], and which could allow us to identify important biological substructures in the network[52,53]. This information could be used, together with the model proposed here, to determine the relevance of such patterns and of the complex interplay between the underlying structure of the network and its functional role, as in the present study.

## Discussion

It is well known that the brain stores information in synaptic conductances, which mediate neural activity and that this in turn affects the birth and death of synapses. To explore what this feedback might entail, we have coupled an auto—associative or attractor neural network model with a model for the evolution of the underlying network topology, which has been used to describe synaptic pruning. Neural network models have long provided a means of relating cognitive processes, such as memory, with biophysical dynamics at the cellular level. This coupled model includes the further ingredient of a changing underlying network structure, in such a way that it can be used as a more general and complete study of synaptic pruning. The intention behind our theoretical framework is to identify a minimal set of ingredients which can give rise to observed phenomena. Specific details of a given system could be added to the model, such as more realistic neuron and synaptic dynamics.

Taken separately, each of the two models involved exhibit continuous phase transitions: between a phase of memory and one of noise in the neural network, and between homogeneous and heterogeneous network topologies in the pruning model. Our coupled model continues to display both of these transitions for certain parameter regimes, but a new discontinuous transition emerges, giving rise to a region of coexistence of phases (also known as hysteresis). In other words, situations with the same parameters but slightly different initial conditions can lead to markedly different outcomes. In this case, whether the attractor neural network is initially capable of memory retrieval influences the emerging network structure, which feeds back into memory

performance. There is therefore a crucial feedback loop between structure and function, which determines the capabilities of the eventual system which the process yields. Our picture thus addresses how neural activity can impact on early brain development, and relates specific dynamic processes in the brain to well-defined mathematical properties, such as bistability and critical-like regions, or the emergence of a feedback loop.

The models we have coupled in this work are the simplest ones able to reproduce the behaviour of interest—namely, associative memory and network topologies with realistic features. However, there are no indications to believe that this feedback will disappear when using more complex models, including the consideration of asymmetric networks, or more realistic choices of the synaptic weights. Hebbian synapses have been considered here as a standard way to define memory attractors, and therefore a useful tool to understand the effect of heterogeneity, and its coupling with memory, on network dynamics. More realistic scenarios could include time dependent synapses, considering for instance learning[43] or fast noise[44], which would indubitably add more complexity to the model. Similarly, particular definitions of the memories could boost the capacity of the network, and even create topologically induced oscillations.

On the other hand, neural systems may not be the only ones to display the properties we found to be sufficient for the existence of this feedback loop between structure and function. Networks of proteins, metabolites or genes also adopt specific configurations (often associated to attractors of some dynamics), and the existence of interactions between nodes might depend on their activity. One may expect that the interplay between form and function we have described in this work could play a role in many natural, complex systems. We have shown some evidence that protein networks seem to have topological features which emerge naturally from the coupled models we have considered here.

Finally, an unresolved question in neuroscience is why brain development proceeds via a severe synaptic pruning—that is, with an initial overgrowth of sypapses, followed by the subsequent atrophy of approximately half of them throughout infancy. Fewer synapses require less metabolic energy, but why not begin with the optimal synaptic density? Navlakha et al. have shown that network properties such as efficiency and robustness can be optimised by a pruning rule which favours short paths[21]. However, in order for synapses to "know" whether they belong to short paths, some kind of back-propagating signal must be postulated. Our coupled model provides a simple demonstration of how network structure can be optimised by pruning, as in Navlakha's model, with a rule that only depends on local information at each synapse—namely, the intensity of electrical current. Moreover, this rule is consistent with empirical results on synaptic growth and death[11–14]. In this view, a neural network would begin life as a more or less random structure with a sufficiently high synaptic density that it is capable of memory performance. This is in keeping with the description given by neuroscientists such as Kolb and Gibb, who say of the initial $10^{14}$ synapses in the human brain: "This enormous number could not possibly be determined by a genetic program, but rather only the general outlines of neural connections in the brain will be genetically predetermined"[54]. Throughout infancy, certain memory patterns are stored, and pruning gradually eliminates synapses experiencing less electrical activity. Eventually, a network architecture emerges which has lower mean synaptic density but is still capable, by virtue of a more optimal structure, of retrieving memories. Moreover, the network structure will be optimised for the specific patterns it stored and, when the emerging networks are scale free, it becomes possible to store orders of magnitude more memory patterns via this mechanism as compared to the networks generated with the uncoupled

(topological) version of the model. This seems consistent with the fact that young children can acquire memory patterns (such as languages or artistic skills) which remain with them indefinitely, yet as adults they struggle to learn new ones[54,55].

## Methods

**The neural network model.** Consider an undirected network with $N$ nodes, and edges which can change in discrete time. At time $t$, the adjacency matrix is $\{e_{ij}(t)\}$, for $i,j = 1,\dots,N$, with elements 1 or 0 according to whether there exists or not an edge between the pair of nodes $(i,j)$, respectively. The degree of node $i$ at time $t$ is $k_i(t) = \sum_j^N e_{ij}(t)$, and the mean degree of the network is $\kappa(t) = N^{-1}\sum_i^N k_i(t)$.

Each node represents a neuron, and each edge a synapse. We shall follow the Amari-Hopfield model, according to which each neuron has two possible states at each time $t$, either firing or silent, given by $s_i(t) = \{0,1\}$[37]. Each edge $(i,j)$ is characterized by its synaptic weight $w_{ij}$, and the local field at neuron $i$ is $h_i(t) = \sum_{j=1}^N w_{ij}e_{ij}(t)s_j(t)$. The states of all neurons are updated in parallel at every time step according to the transition probability $P\{s_i(t+1) = 1\} = \frac{1}{2}[1 + \tanh(\beta[h_i(t) - \theta_i(t)])]$, where $\theta_i(t) = \frac{1}{2}\sum_{j=1}^N w_{ij}e_{ij}(t)$ is a neuron's threshold for firing, and $\beta = T^{-1}$ is a noise parameter controlling stochasticity (analogous to the inverse temperature in statistical physics)[45]. The synaptic weights can be used to encode information in the form of a set of $P$ patterns $\{\xi_i^\mu; \mu = 1,\dots,P\}$, with mean $a_0 = \langle\xi^\mu\rangle$, specific configurations of neural states which can be regarded as memories via the Hebbian learning rule, $w_{ij} = [\kappa_\infty a_0(1-a_0)]^{-1}\sum_{\mu=1}^P(\xi_i^\mu - a_0)(\xi_j^\mu - a_0)$, where $\kappa_\infty = \kappa$ $(t\to\infty)$. The order parameter of the model is the overlap of the state of the neurons with each of the memorized patterns of activity, $m^\mu = (Na_0(1-a_0))^{-1}\sum_{j=1}^N(\xi_i^\mu - a_0)s_i(t)$. The canonical Amari-Hopfield model, which is here a reference, is defined on a fully connected network ($e_{ij} = 1, \forall i \neq j$), and it exhibits a continuous phase transition at the critical value $T = 1$[37].

Notice that, even though synapses are generally asymmetric, we have defined an undirected, symmetric network, in the spirit of previous studies on synaptic pruning[21]. Earlier studies suggest that the inclusion of asymmetry could lead to the induction of chaos, affecting learning, for instance causing the system to oscillate among different states[46]. Here we have decided to simplify the picture and consider symmetric networks, and we expect that, given a reasonable definition of asymmetry, the main results of our work would hold.

**The pruning model.** Edge dynamics is modelled as follows. At each time $t$, each node has a probability $P_i^g(t)$ of being assigned a new edge to another node, randomly chosen. Likewise, each node has a probability $P_j^l(t)$ of losing one of its edges. These probabilities are given by $P_i^g = u(\kappa)\pi(I_i)$ and $P_j^l = d(\kappa)\eta(I_j)$, where we have dropped the time dependence of all variables for clarity, and $I_i$ is a physiological variable that characterizes the local dependence of the probabilities: $I_i = |h_i - \theta_i|$. The first terms on the right-hand side of each equation represent a global dependence to account for the fact that such processes rely in some way on diffusion of different molecules through large areas of tissue, and we take the mean degree $\kappa$ at each time as a proxy. In order to describe synaptic pruning, we choose these probabilities to be consistent with empirical data describing synaptic density in mammals during infancy. The simplest choice is[28,32]

$$u(\kappa) = \frac{n}{N}\left(1 - \frac{\kappa}{2\kappa_\infty}\right), \quad d(\kappa) = \frac{n}{N}\frac{\kappa}{2\kappa_\infty}, \quad (2)$$

where $n$ is the number of edges to be added or removed at each step, which sets the speed of the process, and $\kappa_\infty = \kappa(t\to\infty)$ is the stationary mean degree. It should also be noticed that experimental studies[32] have revealed a fast initial overgrowth of synapses associated with the transient existence of different growth factors. This and other particular mechanisms could be accounted for by adding extra factors in eq. (2). For example, initial overgrowth has been reproduced by adding the term $a\exp(-t/\tau_g)$ in the growth probability[28].

The second terms of the right hand of Eq. (1) were previously considered[28] to be functions of $k_i$. This was meant as a proxy for the empirical observations that synaptic growth and death are determined by neural activity[11–14]. We now make this dependence explicit, and couple the evolving network with the neural dynamics by considering a dependence on the local current at each neuron, $I_i = |h_i - \theta_i|$, as stated before.

**Monte Carlo simulations.** The initial conditions for the neural states are random. For the network topology we can draw an initial degree sequence from some distribution $p(k, t=0)$, and then place edges between nodes $i$ and $j$ with a probability proportional to $k_i(0)k_j(0)$, as in the configuration model. Time evolution is then accomplished in practice via computer simulations as follows. First, the number of links to be created and destroyed is chosen according to two Poisson distributions with means $Nu(\kappa)$ and $Nd(\kappa)$, respectively. Then, as many times as needed according to this draw, we choose a node $i$ with probability $\pi(I_i)$ to be assigned a new edge, to another node randomly chosen; and similarly we choose a new node $j$ according to $\eta(I_j)$ to lose an edge from one of its neighbours, randomly chosen. This procedure uses the BKL algorithm to assure proper evolution towards stationarity[45]. This way, each node can then gain (or lose) an edge via two paths: either through the process with probability $\pi(I_i)$ for a gain (or $\eta(I_i)$ for a loss), or

when it is randomly connected to (or disconnected from) an already chosen node. Therefore, the effective values of the second factors in Eq. 1 are $\tilde{\pi} = 1/2(\pi(I_i) + 1/N)$ and $\tilde{\eta} = 1/2(\eta(I_i) + k_i/(\kappa N))$, where the 1/2 factor is included to assure normalization.

For the sake of simplicity, we shall consider $\tilde{\pi}$ and $\tilde{\eta}$ to be power-law distributed[28], which allows one to move smoothly from a sub-linear to a super-linear dependence with a single parameter, $\tilde{\pi} = I_i^\alpha/(\langle I^\alpha\rangle N)$ and $\tilde{\eta} = I_i^\gamma/(\langle I^\gamma\rangle N)$. This leads to:

$$\pi(I_i) = 2\frac{I_i^\alpha}{\langle I^\alpha\rangle N} - \frac{1}{N}, \quad \eta(I_i, k_i) = 2\frac{I_i^\gamma}{\langle I^\gamma\rangle N} - \frac{k_i}{\kappa N}, \quad (3)$$

where normalization of $\pi, \eta, \tilde{\pi}$ and $\tilde{\eta}$ has been imposed. Notice that by construction the death probability not only depends on $I_i$, but also on the degree $k_i$. Given that, in the memory regime of the Hopfield model, $I_i \propto k_i$, as shown in Results section, here we consider the approximation $\eta \to \eta(I_i)$. Notice also that with this definition the local probabilities could become negative, so we define $\pi(I_i) \to \max(\pi(I_i),0)$ and $\eta(I_i) \to \max(\eta(I_i),0)$. These definitions are most important, as they characterize the coupling between neural activity and structure. However, the particular functions are an arbitrary choice and other ones could be considered. In our scenario, the parameters $\alpha$ and $\gamma$ characterize the dependence of the local probabilities on the local currents and account for the different proteins and factors that control synaptic growth. These could be obtained experimentally, although to the best of our knowledge this has not yet been done.

The timescale for structure changes is set by the parameter $n$ in Eq. 2, whereas the time unit for activity changes, $h_s$, is the number of Monte Carlo Steps (MCS) that the states of all neurons are updated according to the Hopfield dynamics between each structural network update. Our studies show a low dependence on these parameters in the cases of interest, so we only report results here for $h_s = 10$ MCS and $n = 10$.

**The macroscopic state.** The macroscopic state is characterized by the overlap, as defined before, and by the mean neural activity, $M(t) = N^{-1}\sum_{i=1}^N s_i(t)$. Results in the main section are for $P = 1$, so we simplify the notation and use $m = m^1$. A discussion on the effect of learning more patterns is included in subsection Capacity analysis. Also of interest is the degree distribution at each time, $p(k,t)$, whose homogeneity may be measured via $g(t) = \exp(-\sigma^2(t)/\kappa^2(t))$, where $\sigma^2(t)$ is the variance of the distribution and $\kappa$ its mean, as defined before. $g(t) = 1$ for highly homogeneous networks, $p(k) = \delta(k-\kappa)$, and tends to zero for highly heterogeneous ones. Network structure is also characterized by the clustering coefficient for each node, $C_i$, is defined as $C_i = (2t_i)/(k_i(k_i-1))$, where $t_i$ is the number of triangles incident to node $i$, that is, $t_i = 1/2\sum_{j,h} a_{ij}a_{ih}a_{jh}$. We also monitored the Pearson correlation coefficient applied to the edges, $r = ([k_l k_l'] - [k_l]^2)/([k_l^2] - [k_l]^2)$, where $[\cdot] = \frac{1}{\langle k\rangle N}\sum_l(\cdot)$ stands for the average over edges[47], and the time dependence has been dropped for clarity. In this context it can be estimated as $r = \langle k\rangle\langle k^2 k_{nn}(k) - \langle k^2\rangle^2\rangle/(\langle k\rangle\langle k^3\rangle - \langle k^2\rangle^2)$, where $k_{nn}(k)$ is the neighbours mean degree function: $k_{nn,i} = k_i^{-1}\sum_j a_{ij}k_j$[28]. This characterizes the degree–degree correlations, which have important implications for network connectedness and robustness. That is, whereas most social networks are assortative ($r > 0$), almost all other networks, whether biological, technological or information-related, seem to be generically dissassortative ($r < 0$), meaning that high-degree nodes tend to have low-degree neighbours, and vice-versa. Previous studies showed that heterogeneous networks favour the emergence of dissassortative correlations[47–49]. Measures of the global variables on the stationary state are obtained by averaging during a long window of time: $\bar{f} = \Delta t^{-1}\sum_{t=t_0}^{t_0+\Delta t} f(t)$.

**Topological limit.** In the topological limit of the model the topology of the network is independent from its neural state, and one substitutes $I_i \to k_i$, so that $\tilde{\eta}_i = \tilde{\eta}(k_i)$ and $\tilde{\pi}_i = \tilde{\pi}(k_i)$. In this way we can construct a master equation for the evolution of the degree distribution by considering network evolution as a one step process with transition rates $u(\kappa)\tilde{\pi}(k)$ for degree increment and $d(\kappa)\tilde{\eta}(k)$ for the decrement. Approximating the temporal derivative for the expected value of the difference in a given $p(k,t)$ at each time step we get:

$$\frac{dp(k,t)}{dt} = u(\kappa)\tilde{\pi}(k-1)p(k-1,t)$$
$$+ d(\kappa)\tilde{\eta}(k+1)p(k+1,t) \qquad (4)$$
$$- [u(\kappa)\tilde{\pi}(k) + d(\kappa)\tilde{\eta}(k)]p(k,t),$$

which is exact in the limit of no degree–degree correlations between nodes.

Finally, results of the manuscript are for $\gamma = 1$ since, in this topological limit, it corresponds to choosing links at random for removal, given that the probability of choosing an edge $(i,j)$ is then $p_{ij} = \frac{1}{k_i}\tilde{\eta}(k_i) + \frac{1}{k_j}\tilde{\eta}(k_j) = \frac{1}{\langle k\rangle N}$. This can be seen as a first order approximation to the pruning dynamics, and it also induces powerful

simplifications during computations. Furthermore, the relevant parameter determining the behaviour of the system is the ratio between $\alpha$ and $\gamma$, whereas their absolute values only affect quantitatively (see Supplementary Fig. 1).

**Statistics and general methods**. In this work, we used systems sizes $N = 800$, 1600 and 3200, as indicated in each section. Results for $N < 800$ presented strong finite size effects, so they were discarded (data not shown). The sample size for each result was chosen by convergence of the mean value.

**Code availability**. Generated codes are available from the corresponding author upon reasonable request.

**Data availability**. All data that support this study are available from the corresponding author upon reasonable request.

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

## Acknowledgements

We are grateful for financial support from the Spanish MINECO (project of Excellence: FIS2017-84256-P) and from "Obra Social La Caixa".

## Author contributions

A.P.M., J.J.T., J.M. and S.J. designed the analyses, discussed the results and wrote the manuscript. A.P.M. also wrote the codes and performed the simulations.

## Additional information

**Competing interests:** The authors declare no competing interests.

