## [Peer Review File · Nature Communications]

Reviewers' comments:

Reviewer #1 (Remarks to the Author):

The manuscript studies an interplay between structure and function of neural systems in a model of an auto-associative neural network with a stochastic activation and inactivation of synapses. The dynamics of the network activity and synaptic weights, which are related to the memory patterns that should be stored in the network, are the same as in the Amari-Hopfield network. In this work, however, only part of synaptic weights are used for the network dynamics because some of the synapses are constitutively activated or deactivated. How the activation/deactivation probability ratio changes with neural activity is a key model parameter and parameter α controls this in the model – the distribution of active connections to neurons become heterogeneous (i.e., some neurons predominantly have active synapses and some have inactive synapses) if this ratio increases with neural activity ($\alpha > 1$) and the distribution becomes homogeneous if this ratio decreases with neural activity ($\alpha < 1$).

The full model is solved by means of Monte Carlo simulations. Order parameters related to homogeneity of connections g and memory recall quality m are calculated as functions of synaptic activation exponent α and temperature, leading to a phase diagram with four different phases. In line with previous studies it is shown that a heterogeneous phase leads to a better memory performance in noisy networks, compared to a homogeneous phase. A novelty of the new model is the appearance of a first order phase transition and a corresponding bistable phase, in which the actual state of the system depends on the initial conditions. Authors state that this phase is a consequence of the coupled dynamics of structure (synaptic (in)activation) and function (network activity). In order to convince the reader that this is indeed the case it may be beneficial to compare these results with a model with uncoupled dynamics (see comment 2.). I like the simplicity of the proposed model that gives this interesting bistable phase. However, it is not clear if this bistable phenomenon is conceptually different from the bistability resulting from the coupled dynamics between synaptic strength and neural activity (Litwin-Kumar and Doiron, Nat. Commun. 2014; Zenke et al. Nat. Commun. 2015) (see comment 10.). It would be important to articulate what we learn from this work that was not obvious from previous works. Additionally, based on preliminary simulations (results not included in the manuscript) authors mention the possibility of extending the model in order to explain the initial overgrowth of the brain followed by synaptic pruning. Although this could be an interesting point, if presented more clearly, I could not follow the reasoning presented in this part (see comments 6. and 7.).

The manuscript studies an important topic that would be of interest to audiences in wide range of fields, from neuroscience to other biological networks to machine learning. Unfortunately, some of the claims are not strongly supported and/or the underpinning arguments are not clearly presented. In the following I propose a list of questions which I believe should be addressed in a revised version of the manuscript in order to convey a more comprehensive and cohesive message.

Questions and comments:

1. Authors refer to synaptic pruning several times in the manuscript. However, what the model mainly deals with is activation and inactivation of synapses at steady states, suggesting that the model is in fact related to steady-state synaptic turnover.
2. To support the claim that the structure-function interactions really help to retain the bistable memories, it would be useful to compare the phase diagram of the full model (figure 3) with a phase diagram of the corresponding topological limit.
3. Does the phase diagram (figure 3) depend on γ ? Authors assert that qualitative results do not depend on γ and thus present results for the full model with $\gamma = 1$ only. However, the only support for this claim comes from the numerical solutions of the approximate model in the topological limit. The distribution of the number of active synapses may be an important prediction of the model – I would recommend to plot it using the full model in some representative cases.
4. The model is based on an undirected (Amari-Hopfield) network. In contrast, the real neural networks and other biological networks mentioned in the article (i.e. protein interaction,

metabolic) have asymmetric connections. Can we expect the results presented in the article to hold in those cases? Why?

5. How does the capacity of the network depend on the parameters of the model? What is the number of stored patterns P that was used in the study? Is the model behavior invariant to P ?
6. Do results presented in the paper support the idea that it is better to start in an overgrown state and then prune? I do not understand the reasoning presented from line 357 onwards. How does the initial overgrowth help? To achieve the same memory performance, couldn't we have growth and pruning at the same time scales, or predominantly activity-dependent growth with little pruning?
7. Related to the previous point: "...preliminary simulations show that homogeneous IC could strongly improve memory retrieval after pruning." This point is very interesting, unfortunately we do not see the results. Additionally, in line 266 authors suggest that a heterogeneous initial network is effective, but later (270) they say that homogeneous IC could improve memory retrieval. Maybe this is my fault, but I am confused here. The preliminary results discussed here seem to stand in contrast with results presented in the manuscript, which show that heterogeneous IC are advantageous over homogeneous IC.
8. 237–245: "the presence of memory leads to some degree of network heterogenization, which depends on α , giving rise to a memory-heterogeneity feed-back loop". I do not understand that. How does the memory lead to heterogenization? This is not shown in the paper. How do we know whether the "memory-heterogeneity" feedback loop is not just an "activity-structure" feedback loop? Are memories essential? Clarification of the manuscript about this point would be beneficial.
9. The threshold θ for neural activity only depends on synaptic strength but not on the number of inactive synapses. Could this possibly cause a bias in activity that harm memory recall?
10. It was reported that coupled dynamics of synaptic strength and neural activity enables bistable memory (Litwin-Kumar and Doiron, Nat. Commun. 2014; Zenke et al. Nat. Commun. 2015). It would be helpful to explain the conceptual novelty of this work in the context of these references.

Minor errors:

- Seems like the local factor of the gain probability can become negative. Is it not a problem?
- 201: "there is a homogeneous memory phase ..., high heterogeneity g ...". Heterogeneous state has low g , so calling g heterogeneity is quite unfortunate.
- 236: should be $\alpha = 1.5$ (see fig. 2).
- 256: the system first crosses $\alpha_c^t(T)$ and I think that is what author had in mind. Also, the subscript c missing.
- 271: a minus sign in the exponent missing.
- 293: is "contact" a correct synonym of "link" in this abstract context?
- 304: "power-law distributed degree-distributions".
- 306: "related with", should be "related to".
- 326 and 340: relate to.
- 339: "well-defined mathematical properties" – properties of what? I do not understand what authors want to say here.
- The bimodal distribution is not clearly visible in Figure 1a.

Reviewer #2 (Remarks to the Author):

In the paper "The concurrence of form and function in developing networks: An explanation for synaptic pruning" a stochastic Amari-Hopfield model combined with a Barabási-Albert type model for structural plasticity is studied and phase transitions are identified. An interesting result is that memory retrieval seems to be possible under high levels of noise, if high degree nodes are more likely to gain than to lose edges.

While there are interesting points in the paper, I believe it requires a major revision to address the following issues:

1. Lack of sufficient methodological detail. I would have to guess at many places to reproduce the results. Please see questions below.
2. I don't see strong support for a "simple explanation for the existence of an initial explosion of connections followed by a synaptic pruning process", which is one of the major claims (also reflected in the title). If I understand figures 2 & 3 correctly, it is possible to start with a homogeneous or heterogeneous initial state and for sufficiently large α the system will converge to the heterogeneous state and memory recovery is still possible even if the noise increases. Throughout this process the number of connections can stay the same. Maybe an expanded subsection on initial overgrowth (see suggestions) addresses this point.

Questions:

Eq. after 81: What is κ_0 ?

Eq 3: Are there some typos in the left equation? I expected $\pi(I_i) = \frac{2\langle N \langle I^\alpha \rangle \rangle}{\langle I^\alpha \rangle} (I_i^\alpha - \frac{\langle I^\alpha \rangle}{2})$

Eq 3: It is not clear to me how the second equality is derived.

134 - 135: What do you mean with "the number h_s of MCS between each configuration updating"? Is this the number of times the states of all neurons are updated according to the equation after line 76 in between each structural update step?

143: What do you mean with "the Pearson correlation coefficient applied to the edges"? Naively I expected $\langle (k_i - k)(k_j - k) \rangle / \sigma_k^2$ instead of the (somewhat complicated) formula for r .

144: How is the function $k_{nn}(k)$ defined?

154: "This immediately leads ..." I did not see how this master equation follows immediately. Can you explain a bit or give a reference? I guess there are also typos: $dp(k,t) = \dots$ and $d(\kappa)\tilde{\eta}(k)$.

172: "Notice that the condition ..." Can you elaborate on this? E.g. how do we see that it is mandatory and why do you find the opposite of [17]?

Figure 1b: $\eta_i \propto k_i$ and $\pi_i \propto k_i^\alpha$ is for the green curve? What was the choice for the purple condition?

Figure 1c: What is the normalized local field here? On line 74 the local field is defined as h_i , but h_i can also be negative and it is not constant (depends on s_j).

Figure 2: How is m defined? Is it $1/P \sum_{\mu} \langle m^{\mu} \rangle$ where the expectation is over time? How many patterns P are stored and recalled? Is $\kappa_m = \kappa_{\infty}$?

Figure 3: What do you exactly mean by "obtained from analysis of the control parameters"? How do the transition lines depend on the parameters, e.g. κ_{∞} , n , P , a_0 ? Is there an intuitive explanation for why a larger α allows recall even under noise levels above $T > 1$?

296, 302: How is the clustering coefficient defined and how did you find in the model a decay as k^{-1} ?

298: Is the nearest neighbour degree function K the same as k_{nn} defined on page 5? If yes, please use the same notation.

369: "the probabilities of birth and death of synapses ... should be approximately linear". I guess this implies $\alpha \approx \gamma \approx 1$. But on line 183 you argue that only the relation between α and γ matters, which I interpret as $\alpha \approx \gamma$. In this light, the statement in the discussion is not as general as it could be.

Suggestions:

Eq 1: The definition of I_i follows many lines later. I think it would help the reader to bring it closer to Eq 1.

Eq 1 & 3: It could be nice to have the tilde already in Eq 1 and say somewhere that $\sum_i \pi(I_i) = 1$.

Table of page 6: $\tilde{\eta}(k_i) > \tilde{\pi}(k_i)$ for $k_i > k$ etc.

183: The relevant parameter is the ratio α/γ .

Paragraph: Synaptic pruning with initial overgrowth: This looks interesting. I think the paper would profit from expanding this part and having more than preliminary simulations.

As they are now, the paragraphs on initial overgrowth and protein interactions feel more like discussion paragraphs than results. Maybe it would be worthwhile to move them to the discussion section.

A somewhat peculiar assumption of the model is that the connection between two neurons i and j is either w_{ij} given by the Hopfield prescription or 0 (when $e_{ij} = 0$), i.e. as soon as a synapse appears between these two neurons its strength is set to a constant value and now growth phase takes place. Maybe you could comment on why this is a reasonable assumption.

63: underlying dynamics of

I am not an expert on structural plasticity or synaptic pruning, but there seem to be many, also recent, interesting experimental and theoretical studies on this topic that could be worth citing:

- Anthony Holtmaat and Karel Svoboda have done interesting experimental studies, e.g. <http://dx.doi.org/10.1016/j.neuron.2005.01.003> or <http://dx.doi.org/10.1038/nrn2721>
- On the theory side e.g. <http://dx.doi.org/10.1371/journal.pcbi.1004347>, <http://dx.doi.org/10.1371/journal.pcbi.1002689> or <https://arxiv.org/abs/1609.05730>

Reviewer #1 (Remarks to the Author):

The manuscript studies an interplay between structure and function of neural systems in a model of an auto-associative neural network with a stochastic activation and inactivation of synapses. The dynamics of the network activity and synaptic weights, which are related to the memory patterns that should be stored in the network, are the same as in the Amari-Hopfield network. In this work, however, only part of synaptic weights are used for the network dynamics because some of the synapses are constitutively activated or deactivated. How the activation/deactivation probability ratio changes with neural activity is a key model parameter and parameter α controls this in the model – the distribution of active connections to neurons become heterogeneous (i.e., some neurons predominantly have active synapses and some have inactive synapses) if this ratio increases with neural activity ($\alpha > 1$) and the distribution becomes homogeneous if this ratio decreases with neural activity ($\alpha < 1$). The full model is solved by means of Monte Carlo simulations. Order parameters related to homogeneity of connections g and memory recall quality m are calculated as functions of synaptic activation exponent α and temperature, leading to a phase diagram with four different phases. In line with previous studies it is shown that a heterogeneous phase leads to a better memory performance in noisy networks, compared to a homogeneous phase. A novelty of the new model is the appearance of a first order phase transition and a corresponding bistable phase, in which the actual state of the system depends on the initial conditions. Authors state that this phase is a consequence of the coupled dynamics of structure (synaptic (in)activation) and function (network activity). In order to convince the reader that this is indeed the case it may be beneficial to compare these results with a model with uncoupled dynamics (see comment 2.). I like the simplicity of the proposed model that gives this interesting bistable phase. However, it is not clear if this bistable phenomenon is conceptually different from the bistability resulting from the coupled dynamics between synaptic strength and neural activity (Litwin-Kumar and Doiron, Nat. Commun. 2014; Zenke et al. Nat. Commun. 2015) (see comment 10.). It would be important to articulate what we learn from this work that was not obvious from previous works. Additionally, based on preliminary simulations (results not included in the manuscript) authors mention the possibility of extending the model in order to explain the initial overgrowth of the brain followed by synaptic

pruning. Although this could be an interesting point, if presented more clearly, I could not follow the reasoning presented in this part (see comments 6. and 7.).

The manuscript studies an important topic that would be of interest to audiences in wide range of fields, from neuroscience to other biological networks to machine learning. Unfortunately, some of the claims are not strongly supported and/or the underpinning arguments are not clearly presented. In the following I propose a list of questions which I believe should be addressed in a revised version of the manuscript in order to convey a more comprehensive and cohesive message.

We thank the referee for a very careful reading of the manuscript and for the comments and suggestions. We very much appreciated the thorough review. We include below our responses to the issues listed as “Questions and comments” and “Minor errors” in the report, indicating in each question which changes have been made in the manuscript. Also, all relevant new or re-structured text in the manuscript have been printed in blue – this will not appear in the final version of the text.

We have changed the layout of the manuscript to fit the editorial policies of the journal Nature Communication. As a consequence, the former “model” section has been moved to the methods (starting pg 8), and we have included a light summary of the main points of the model as a subsection in the results (“Model construction”, pg 2).

The suggestions made in the reviews have also let us to introduce two new subsections in the results: a first one leading with the experimental plausibility of the model (“Synaptic pruning”, starting pg 3), and a second one analyzing the capacity of the model (“Capacity analysis”, starting pg 6). Due to these changes, we have also re-structured slightly the main layout of the results section, which is now divided into subsections. Finally, we have also added two appendices of supplementary information: one analyzing the role of γ and other parameters in the model (“Parameter analysis”, pg 13), and another showing the phase diagram of the topological limit. Here we also show some representative examples of the stationary degree distribution of the system in the full model (“Topological limit”, pg 13).

We expect that this changes are sufficient to support our claim of the relevance of this structure-function feed-back loop in synaptic pruning, and that the new text provides a more coherent explanation.

Questions and comments:

1. Authors refer to synaptic pruning several times in the manuscript. However, what the model mainly deals with is activation and inactivation of synapses at steady states, suggesting that the model is in fact related to steady-state synaptic turnover.

We agree with the reviewer in that the relation of our model with synaptic pruning was not made explicit enough in the manuscript. In order to fix this problem, we have included a section dealing in more detail with the temporal evolution of the mean connectivity in our model, which is inspired by experimental results [1,2]. In particular, in the novel section “Synaptic pruning” (starting in pg 3, line 194) we fit two experimental data-sets on neuron connectivity during infancy. These data-sets correspond to two independent studies of very different nature. The first one [1] comprises data from a 2015 electron microscopy imaging study on the mouse somatosensory cortex, whereas the second one [2] is from a 1997 work on the human infant brain, in which connectivity data is obtained by directly counting synapses in tissues from autopsies. Regardless of this, both data-sets show an initial growth of the connectivity followed by the pruning process. Our model provides an acceptable approximation of the pruning section of both data-sets, given the right set of parameters. Details have been included in the manuscript, the data and theoretical fit are shown in figure 2 (pg 4). Deviations from the theoretical result should not be surprising, since our model does not consider any details of the neurons, brain region or even species. But, still, both data-sets follow a similar mean tendency that we can reproduce.

We also mention that, in order to fit the initial growth and maximum of the connectivity, we have needed to include a time-dependent, non-linear growing factor in the probabilities, as it was already discussed in [3]. This takes into account the transient existence of some remaining growth factors from the pre-natal period. With this consideration, the initial growth and maximum in the connectivity can also be fit, as shown in the bottom graphs. However, this adds new complexity into the model and, since we were mainly interested in characterizing first the effect of the coupling between structure and physiology, it was not taken into consideration in the rest of the manuscript.

We expect that these additions will convince the reader that the model is in fact dealing with synaptic pruning during development, and not only with the stationary state of the network.

[1] P R Huttenlocher and A S Dabholkar. "Regional differences in synaptogenesis in human cerebral cortex." *Journal of Comparative Neurology*, 387:167, 1997.

[2] S Navlakha, A L Barth, and Z Bar-Joseph. "Decreasing-rate pruning optimizes the construction of efficient and robust distributed networks." *PLoS Comput Biol*, 11:e1004347, 2015.

[3] S Johnson, J Marro, and J J Torres. "Evolving networks and the development of neural systems." *Journal of Statistical Mechanics: Theory and Experiment*, P03003, 2010.

2. To support the claim that the structure-function interactions really help to retain the bistable memories, it would be useful to compare the phase diagram of the full model (figure 3) with a phase diagram of the corresponding topological limit.

We thank the referee for pointing out the convenience of this analysis. We have included a section of supplementary information ("Topological limit", starting in pg 13, line 1010) with the corresponding diagram (figure 8) and a consequent discussion to clarify this matter. We have also included a comment in the main text (pg 5, starting line 340) referring to this.

With this new diagram, we prove that, without the physiology-structure coupling, network structure is simply determined by α and, this, in turns, characterizes the memory transition, according to previous studies [1].

In this way, for $\alpha < 1$, networks are homogeneous, and there is a continuous transition from memory to noise. The critical temperature for this transition moves from $T_C = 1$ for completely homogeneous networks (so that $p(k) = \delta_{k,k_0}$) for $\alpha \ll 1$, to higher values as the nodes degrees gain some heterogeneity, according to previous studies [1].

For $\alpha > 1$ and a finite-size system ($N=800$ in the results shown), the temperature of the transition keeps growing with α . In the thermodynamic limit, this takes place an infinite temperature according to the literature [2]

As a result of the lack of feed-back from the physiology, the bistability region disappears, since the structure of the network does no longer depend on the memory state.

- [1] JJ Torres, MA Munoz, J Marro and PL Garrido. "Influence of topology on the performance of a neural network." *Neurocomputing*, 58, 229, 2004.
- [2] M Leone et al. "Ferromagnetic ordering in graphs with arbitrary degree distribution." *The European Physical Journal B-Condensed Matter and Complex Systems* 28:2, 2002.

3. Does the phase diagram (figure 3) depend on γ ? Authors assert that qualitative results do not depend on γ and thus present results for the full mode with $\gamma = 1$ only. However, the only support for this claim comes from the numerical solutions of the approximate model in the topological limit. The distribution of the number of active synapses may be an important prediction of the model – I would recommend to plot it using the full model in some representative cases.

We thank the reviewer for suggesting this analysis, which has been included as supplementary information ("Parameter analysis", starting pg 13, line 978). We show an analysis of the phase diagram of the system for different values of γ in figure 7. This shows that the bistability phase moves with γ , as it is expected from results in the topological limit, but qualitative results remain unchanged. A complete description of the diagrams and this phenomenon has also been included in the SI, and we have also included two comments on this in the results section (starting pg 4, line 263) and the methods (pg 10, line 781, respectively).

We have also included a graph of the degree distribution $p(k)$ on different characteristic scenarios also in figure 7. First, in the main plot, we show $p(k)$ for a (T, α) point in each of the phases: homogeneous memory, homogeneous noise and heterogeneous memory. The homogeneous distributions are fairly similar, whereas the heterogeneous one is bimodal. Inset 1 of this graph shows a comparison between homogeneous and heterogeneous IC in the bistable region: homogeneous IC fall into the noisy homogeneous phase, whereas heterogeneous ones maintain memory and organize into a bimodal distribution.

Finally, inset Fig. 2 shows a comparison of the degree distributions obtained for different values of γ , for $\alpha = \gamma$ and $T = 0.5$, which are near the transition line $\alpha_c(T)$. These show that the value of γ does not affect decisively the decay of the degree distribution.

4.- The model is based on an undirected (Amari-Hopfield) network. In contrast, the real neural networks and other biological networks mentioned in the article (i.e. protein interaction, metabolic) have asymmetric connections. Can we expect the results presented in the article to hold in those cases? Why?

We expect from earlier studies in asymmetric Hopfield networks [1], that the consideration of asymmetric synapses could lead to the induction of chaos or other dynamical behaviors affecting learning. However, the actual dynamics of the network would depend on the particular definition of the connections. For instance, in [2] synaptic weights are defined as $w_{ij} \propto w_{ij}^{\text{Hop}} + \sum_{\mu} (\xi_i^{\mu} \times \xi_j^{\mu+1})$, so that they induce learning of a sequence of patterns, and oscillations amongst them. On the other hand, in [3,4] the asymmetry does not lead to new dynamical phases, but acts as a source of noise, creating temporal aperiodic small oscillations of the overlap in the traditional memory state, as a consequence of inducing a change in the dynamical structure of the memory attractors.

Therefore, the definition of the asymmetry itself would be a new – and critical – assumption of the model. We have hence decided to simplify the picture considering symmetric networks as it is also commonly done in autoassociative neural network studies [5] or evolutionary models for proteins interaction networks [6]. We expect that, given a reasonable definition of the synaptic weights, the main results of our work would hold – such as the appearance of a feed-back loop and bistability region – when asymmetric networks are considered.

We do consider this is an interesting point that should be further valued in future studies, but that is beyond the scope of this work. Therefore, we have included a discussion of this in the manuscript in the definition of the model in the methods (starting pg 9, line 625) and in the discussion (starting pg 8, line 539), since we consider this an interesting point to be valued in future works.

- [1] Sompolinsky, Haim, and I. Kanter. "Temporal association in asymmetric neural networks." *Physical review letters* 57.22 (1986): 2861.
- [2] Riedel, U., R. Kühn, and J. L. Van Hemmen. "Temporal sequences and chaos in neural nets." *Physical review A* 38.2 (1988): 1105.
- [3] Zanette, D. H., and A. S. Mikhailov. "Mutual synchronization in ensembles of globally coupled neural networks." *Physical Review E* 58.1 (1998): 872.
- [4] Derrida, Bernard, Elizabeth Gardner, and Anne Zippelius. "An exactly solvable asymmetric neural network model." *EPL (Europhysics Letters)* 4.2 (1987): 167.
- [5] Navlakha, Saket, Alison L. Barth, and Ziv Bar-Joseph. "Decreasing-rate pruning optimizes the construction of efficient and robust distributed networks." *PLoS computational biology* 11.7 (2015): e1004347.
- [6] Berg, Johannes, Michael Lässig, and Andreas Wagner. "Structure and evolution of protein interaction networks: a statistical model for link dynamics and gene duplications." *BMC evolutionary biology* 4.1 (2004): 51.

5. How does the capacity of the network depend on the parameters of the model? What is the number of stored patterns P that was used in the study? Is the model behavior invariant to P ?

This is indeed an interesting question that arises a complex topic. In fact, in order to answer this concern and a similar one arose by another review, we have included a novel subsection in the results: "Capacity analysis", starting pg 6, line 406.

As in other neural network models, our results hold for a small number of patterns, whereas when this number is increased the system is trapped in the spin glass state (SG), in which the network is not able to distinguish amongst some of the memorized patterns. In particular, our results agree with studies in other heterogeneous networks considering a hebbian learning [1], that find a fast decrease in network capacity with heterogeneity. This result is shown in figure 5a, pg 6.

As an expansion of this work, we have proposed a localized definition of the patterns motivated by physiological studies [2], and showed that network capacity is boosted in this manner. In this picture, each pattern is associated with a "patch" of active neurons, whereas the rest of the network keeps silent, thus reducing the mean network activity associated with a memory state. We have included in the manuscript a discussion on

the capacity curves of the system for this case, and showed that the number of memorized patterns compatible with memory can be dramatically increased in this manner. We also find a great dependence on α , as expected. For $\alpha < 1$, memory is preserved only for small P . However, in the heterogeneous phase, the competition between the different patterns boosts network heterogeneity, and the capacity of the network increases greatly, as it is seen in the new pictures.

As for the dependence in the other parameters of the model, we have found that structural parameters such as n or h_s do not have great influence on the dynamics of the network, as long as they are kept within reasonable values. The same holds for κ_0 and κ_∞ . We have included a section of supplementary information ("Parameter Analysis", starting pg 13, line 978) of the paper addressing this question.

Finally, we have also included a paragraph on this results in the results section, pg 4 starting line 263, and in the discussion, pg 8 starting line 548.

[1] L G Morelli and M N Kuperman. "Associative memory on a small-world neural network." *The European Physical Journal B-Condensed Matter and Complex Systems* 38:3, 2004.

[2] T Akam and D M Kullmann. "Oscillatory multiplexing of population codes for selective communication in the mammalian brain." *Nat. Revs. Neuroscience* 15:2, (2014).

6. Do results presented in the paper support the idea that it is better to start in an overgrown state and then prune? I do not understand the reasoning presented from line 357 onwards. How does the initial overgrowth help? To achieve the same memory performance, couldn't we have growth and pruning at the same time scales, or predominantly activity-dependent growth with little pruning?

Results in the manuscript state that heterogeneous initial conditions help promoting memory by providing a better configuration for memory at the onset of pruning. What we argue in the discussion of the manuscript is that this same result could be achieved with homogeneous IC via an initial overgrowth, during which the microscopical rules for synapse birth and death would be as defined before. Our understanding is that this would create a long transient of increased connectivity, which stabilizes memory even for high noise and homogeneous networks. Due to the underlying topological dynamics, this allows the network to become more

heterogeneous before the actual pruning begins, thus it would have a similar effect to considering heterogeneous IC. However, the former initial overgrowth is more realistic to happen in nature, as experimental pruning curves have shown. We discuss this in detail in section “Synaptic pruning”, starting pg 3, line 194 of the revised manuscript, where we have included a fit of experimental data on synaptic pruning .

On the other hand, synaptic pruning is needed to decrease energy consumption, as previous studies have indicated [1,2]. Even though high stationary connectivity would improve memory in a diluted network, this is not seen in nature, and also it does not allow for the formation of highly heterogeneous structures (since every node has relatively high -and thus similar - connectivity).

We agree with the referee in that this was not clearly stated in the discussion before. Also, we have moved the paragraph on initial overgrowth from the main text to the discussion section (starting pg 8, line 571 of the new manuscript), and rewritten it so that it does not induce doubt and the arguments are presented more clearly.

[1] S Navlakha, A L Barth, and Z Bar-Joseph. "Decreasing-rate pruning optimizes the construction of efficient and robust distributed networks." PLoS Comput Biol, 11:e1004347, 2015. of Neurophysiology, 44:247, 1980.

[2] E Frank. "Synapse elimination: For nerves it's all or nothing." Science, 275:324, 1997.

7. Related to the previous point: “...preliminary simulations show that homogeneous IC could strongly improve memory retrieval after pruning.” This point is very interesting, unfortunately we do not see the results. Additionally, in line 266 authors suggest that a heterogeneous initial network is effective, but later (270) they say that homogeneous IC could improve memory retrieval. Maybe this is my fault, but I am confused here. The preliminary results discussed here seem to stand in contrast with results presented in the manuscript, which show that heterogeneous IC are advantageous over homogeneous IC.

We agree with your concern in this point, since this is, in fact, a very unfortunate typo in the writing, which we have fixed. As the referee suggests, it was meant to say “...preliminary simulations show that **heterogeneous** IC could strongly improve memory retrieval after pruning.”, in accordance with the rest of the discussion. We have

consequently fixed this mistake. Notice that this paragraph has been now moved to the discussion (pg 8, starting line 571), in relation to the previous question.

8. 237–245: “the presence of memory leads to some degree of network heterogenization, which depends on α , giving rise to a memory-heterogeneity feed-back loop”. I do not understand that. How does the memory lead to heterogenization? This is not shown in the paper. How do we know whether the “memory-heterogeneity” feedback loop is not just an “activity-structure” feedback loop? Are memories essential? Clarification of the manuscript about this point would be beneficial.

In our model, the structure of the network is driven by the physiological state through the current $I_i = |h_i - \theta_i|$, which characterizes the local probabilities of edge birth and death. In order to allow for a heterogeneous structure, an ordered activity of the system is needed, and this means the formation of memories. This is so since only when the system is in a memory state there is a direct correlation between I_i and k_i , as shown in figure 1b of the manuscript. This allows for the emergence of structure – as given by the local probabilities (which, in our case, are characterized by α). On the contrary, if the network is in a noisy state, edge birth and death would be a random process, and thus lead to a homogeneous network configuration.

We have included this discussion in the result section of the manuscript (pg 5, starting line 347).

9. The threshold θ for neural activity only depends on synaptic strength but not on the number of inactive synapses. Could this possibly cause a bias in activity that harm memory recall?

We thank the referee for pointing this out. This is a typo in the definition of the thresholds, which should read “ $\theta_i = 1/2 \sum_{j=1}^N e_{ij} w_{ij}$ ”, so that it does take into account only the existing synapses. We have fixed this mistake in the text (pg 8, line 611).

10. It was reported that coupled dynamics of synaptic strength and neural activity enables bistable memory (Litwin-Kumar and Doiron, Nat. Commun. 2014; Zenke et al. Nat. Commun. 2015). It would be helpful to explain the conceptual novelty of this work in the context of these references.

We thank the referee for pointing out these works, which we have included as references in the introduction of the manuscript (pg 2, line 87). After a careful reading, we found they provide an interesting insight into the role of bistability in learning. However, from our understanding both the subject of the studies and the means that give rise to the bistability are very different from our present work. In fact, bistability in these works arises from the competition between different sources of synaptic depression and facilitation, whereas in our work it is due to the coupling between structure and physiology. Also, in these works a fixed network topology is assumed, whereas in our model it is its dependence on the currents I_i what gives rise to the bistability.

Minor errors:

Seems like the local factor of the gain probability can become negative. Is it not a problem?

We thank the referee for pointing this out, as it is a typo due to an oversimplification on the writing. We have rewritten it as $\pi_i \rightarrow \max\{0, \pi_i\}$, which is what we actually use in the model (see pg 9, starting line 709). Also, there was a typo in the normalization of π

and in the writing of η in equation 3, which has now been addressed, and we have explicitly included the condition that π , η , $\widetilde{\eta}$ and $\widetilde{\pi}$ are to be normalized (lines 696 and 704, pg 9).

- 201: “there is a homogeneous memory phase ..., high heterogeneity g ...”. Heterogeneous state has low g , so calling g heterogeneity is quite unfortunate.

We agree with the reviewer that this can lead to some confusion. Therefore, we have redefined g as the “homogeneity” parameter.

236: should be $\alpha = 1.5$ (see fig. 2).

Typo fixed.

- 256: the system first crosses $\alpha_c t(T)$ and I think that is what author had in mind. Also, the subscript c missing.

Correction made (now in line 381, pg 6).

271: a minus sign in the exponent missing.

Typo fixed (this paragraph is now in the discussion, line 578, pg 8).

- 293: is “contact” a correct synonym of “link” in this abstract context?

We agree that this was a poor choice of words. Therefore, we have rewritten this sentence as “To illustrate in more detail the relation with the previous works, ...”, in line 453, pg 7.

- 304: “power-law distributed degree-distributions”.

This paragraph has been changed in the new manuscript and we have removed this sentence (starting line 4466, pg 7).

306: “related with”, should be “related to”.

Typo fixed.

326 and 340: relate to.

Typo fixed.

- 339: “well-defined mathematical properties” – properties of what? I do not understand what authors want to say here.

We agree with the reviewer that this should be better explained. By “well-defined mathematical properties” we referred to the existence of phase

transitions, biestability or the feed-back loop between form and function, which are well characterized mathematically.

With this sentence we relate We have rewritten this paragraph to specify more clearly what we mean (line 525, pg 8 of the new manuscript).

- The bimodal distribution is not clearly visible in Figure 1a.

We have decreased the bin spacing in the plot, so that more points of the bimodal distribution are visible.

Reviewer #2 (Remarks to the Author):

In the paper "The concurrence of form and function in developing networks: An explanation for synaptic pruning" a stochastic Amari-Hopfield model combined with a Barabási-Albert type model for structural plasticity is studied and phase transitions are identified. An interesting result is that memory retrieval seems to be possible under high levels of noise, if high degree nodes are more likely to gain than to lose edges.

While there are interesting points in the paper, I believe it requires a major revision to address the following issues:

1. Lack of sufficient methodological detail. I would have to guess at many places to reproduce the results. Please see questions below.
2. I don't see strong support for a "simple explanation for the existence of an initial explosion of connections followed by a synaptic pruning process", which is one of the major claims (also reflected in the title). If I understand figures 2 & 3 correctly, it is possible to start with a homogeneous or heterogeneous initial state and for sufficiently large α the system will converge to the heterogeneous state and memory recovery is still possible even if the noise increases. Throughout this process the number of connections can stay the same. Maybe an expanded subsection on initial overgrowth (see suggestions) addresses this point.

We thank the reviewer for his/her thorough review and highly appreciate the comments and suggestions, which significantly contributed to improving the quality of the publication. Please find below a detailed response to each of the comments. We indicate in each question which changes have been made in the manuscript, and, also, all new or re-structured text in the manuscript have been printed in blue – this will not appear in the final version of the text.

We have changed the layout of the manuscript to fit the editorial policies of the journal Nature Communication. As a consequence, the former "model" section has been moved to the methods (starting pg 8), and we have included a light summary of the main points of the model as a subsection in the results ("Model construction", pg 2).

The suggestions made in the reviews have also let us to introduce two new subsections in the results: a first one leading with the experimental plausibility of the model ("Synaptic pruning", starting pg 3), and a second one analyzing the capacity of the model ("Capacity analysis", starting pg 6). Due to these changes, we have also re-structured slightly the main

layout of the results section, which is now divided into subsections. Finally, we have also added two appendices of supplementary information: one analyzing the role of γ and other parameters in the model (“Parameter analysis”, pg 13), and another showing the phase diagram of the topological limit. Here we also show some representative examples of the stationary degree distribution of the system in the full model (“Topological limit”, pg 13).

Regarding the second issue arose by the reviewer, we have considered expanding the subsection on initial overgrowth, as suggested, but we understand that a comprehensive analysis of this model that fully characterizes its dynamics is beyond the scope of this study. Here we are mainly concern with the characterization of the coupling between memory and structure, and with the implications of this feed-back loop. Therefore, whereas we agree that this would be a very interesting result to unfold in future works, we have followed the suggestion of the reviewer and moved the paragraph on initial overgrowth to the discussion section.

We expect that the rest of the changes we have made will support our claim of the relevance of this structure-function feed-back loop in understanding synaptic pruning, and that the new text provides a more coherent and comprehensive explanation.

Questions:

Eq. after 81: What is κ_0 ?

κ_0 is the mean connectivity at $t=0$: $\kappa_0 = \kappa(t=0)$. We have included the definition of $\kappa_0 = \kappa(t=0)$ in line 620, pg 8.

Eq 3: Are there some typos in the left equation? I expected $\pi(I_i) = \frac{2\{N\}}{(I_i^\alpha - \frac{\{ \}}{\{2\}})$

Yes, there was a typo in the normalization of $\pi(I_i)$. We have fixed it and rewritten the equation in a more straight forward manner as $\pi(I_i) = 2 \frac{I_i^\alpha}{\langle I^\alpha \rangle N} - 1/N$ (see line 704 on the manuscript, pg 9). There was also a typo in the writing of η , where a factor was missing, which has also been addressed (see next question).

Eq 3: It is not clear to me how the second equality is derived.

There was a mistake in the writing of this equation, in which a second term is missing. This has been fixed. The expression is obtained by defining $\tilde{\eta} = I_i^\gamma / (\langle I_i^\gamma \rangle N)$, and taking into account that $\tilde{\eta} = \frac{1}{2} (\eta + k_i / (\kappa N))$, where the $\frac{1}{2}$ factor is to assure normalization. From this we get $\eta = 2 I_i^\gamma / (\langle I_i^\gamma \rangle N) - k_i / (\kappa N)$ as equation 3 reads now (line 695, pg 9 of the new manuscript).

134 - 135: What do you mean with "the number h_s of MCS between each configuration updating"? Is this the number of times the states of all neurons are updated according to the equation after line 76 in between each structural update step?

We meant exactly as the reviewer proposes, so we have included this suggestion to make the definition more clear (line 722, pg 10) .

143: What do you mean with "the Pearson correlation coefficient applied to the edges"? Naively I expected $\langle (k_i - k)(k_j - k) \rangle / \sigma_k^2$ instead of the (somewhat complicated) formula for r .

Here we follow the definition of the Pearson correlation coefficient given in [1] and also derived in [2] for an evolving network similar to ours. We note that the definition the reviewer proposes calculates the correlation between the degrees of every pair of nodes, whereas we apply it to every edge - thus, only considering connected pairs. We have included a

derivation of the used formula from the general one in the paper (starting line 743, pg 10).

[1] Newman M E J 2003 The structure and function of complex networks SIAM Rev. 45 167

[2] S Johnson, J Marro, and J J Torres. "Evolving networks and the development of neural systems." Journal of Statistical Mechanics: Theory and Experiment, P03003, 2010.

144: How is the function $k_{nn}(k)$ defined?

The function $k_{nn}(k)$ gives the mean degree of the nearest neighbors of nodes with degree k . We have included the definition of the function $k_{nn}(k)$ in the text in line 750, pg 10.

154: "This immediately leads ..." I did not see how this master equation follows immediately. Can you explain a bit or give a reference? I guess there are also typos: $dp(k,t) = \dots$ and $d(\kappa)\tilde{\eta}(k)$.

We followed the reviewer suggestion and added a more detailed argument on how to obtain the master equation, starting line 761, pg 10 (notice that this derivation has been moved to the methods section in the new manuscript). Roughly, we consider the evolution of the network as a one step process with transition rates $u(\kappa) \tilde{\pi}(k)$ for an increment in the degree and $d(\kappa) \tilde{\sigma}(k)$ for a decrement. We then approximate the expected value of the difference in a given $p(k,t)$ at each time step as the temporal derivative, and from it we derive the master equation. There were also typos in the equation, which have been corrected.

We expect that with the included discussion, the meaning and derivation of the equation will be more clear to follow.

172: "Notice that the condition ..." Can you elaborate on this? E.g. how do we see that it is mandatory and why do you find the opposite of [17]?

This refers to the condition $\tilde{\pi} = \tilde{\eta}$ for high k_i . We find that this condition is mandatory since otherwise high-degree nodes would continue to gain links if $\tilde{\pi} > \tilde{\eta}$, or would lose them and a cut-off would emerge if $\tilde{\pi} < \tilde{\eta}$. This is shown in figure 1b of the original manuscript (or figure 2b of the new version). Here the green curve shows $g(\alpha)$ for a choice of local probabilities that do not fulfill $\tilde{\pi} = \tilde{\eta}$. In particular,

we took $\eta \propto k_i$ and $\pi_i \propto k_i^\alpha$, in which case $\widetilde{\eta} = \frac{1}{2} (k_i / (\langle k \rangle N) + 1/N)$ and $\widetilde{\pi} = \frac{1}{2} (k_i^\alpha / \langle k^\alpha \rangle N) + k_i / (\langle k \rangle N)$. Notice that with this choice there is no value of α for which the condition $\widetilde{\pi} = \widetilde{\eta}$ is true for every $k \gg 1$. Therefore, in this case the transition is discontinuous, as shown in the plot, and it does not go through a critical point.

On the other hand, for the probabilities indicated in eq3 of the new manuscript, we have trivially that $\widetilde{\pi} = \widetilde{\eta}$, and the transition between homogeneous and heterogeneous networks is continuous, as shown in figure 2b of the new version by the purple curve.

We think this was not as clearly stated in the previous version as it should have been, so it has been rewritten in more straight-forward manner in pg 3, starting line 184. This is in relation also with the previous and next questions, which arose doubts about the same section.

Figure 1b: $\eta_i \propto k_i$ and $\pi_i \propto k_i^\alpha$ is for the green curve? What was the choice for the purple condition?

In figure 1b, we chose $\eta_i \propto k_i$ and $\pi_i \propto k_i^\alpha$ for the purple curve, since these probabilities do not meet the condition $\widetilde{\eta}_i = \widetilde{\pi}_i$, and therefore lead to a discontinuous transition. On the other hand, the green curve is for $\eta \propto k_i$ and $\pi \propto k_i - \kappa(t)$.

We have made these definitions explicit in the main text (starting line 184, pg 3) and in the caption of figure 1b (pg 3). We have also rewritten the discussion on the local probabilities (starting line 761, pg 10) to clarify this argument. In short, in the topological limit one needs to assure that $\eta_i = \pi_i$ is exactly fulfilled for $k \gg 1$ in order to have a continuous transition.

Figure 1c: What is the normalized local field here? On line 74 the local field is defined as h_i , but h_i can also be negative and it is not constant (depends on s_j).

We thank the referee for pointing this out. There is a mistake in the nomenclature, which should be “normalized local current”, and its defined for each node as $I_i / (\langle I \rangle N)$. That is, the incoming current at each node normalized by the “total” current in the network. We have fixed the nomenclature and added this definition in the caption of figure 1c (pg 3).

Figure 2: How is m defined? Is it $1/P \sum_{\mu} \langle m^{\mu} \rangle$ where the expectation is over time? How many patterns P are stored and recalled? Is $\kappa_m = \kappa_{\infty}$?

Results shown in the paper are for $P=1$, and we define $m = \langle m^1 \rangle$, where the expectation is over time once the network is in the stationary state. This definition has been included in the manuscript, in the model definition, in line 734, pg 10. We have also included a novel section in the results discussing the effect of increasing the number of memorized patterns (that is section “Capacity analysis”, starting pg 6, line 406). We have found that, as in other neural network models, results hold for a small number of patterns, whereas when this number is increased the system is trapped in the spin glass state (SG). Our results hold with capacity analysis in other heterogeneous networks using a hebbian learning [1], that find a fast decrease in network capacity with heterogeneity. This results are shown in figure 5a (pg 6) of the revised manuscript.

As an expansion of this work, we have proposed a localized definition of the patterns motivated by physiological studies [2], and showed that network capacity is boosted in this manner. In this picture, each pattern is associated with a “patch” of active neurons, whereas the rest of the network keeps silent, thus reducing the mean network activity associated with a memory state. We have included in the manuscript the capacity curves of the system for this case as well (figure 5b, pg 6), and showed that the number of memorized patterns compatible with memory can be dramatically increased in this manner. We also find a great dependence on α , as expected. For $\alpha < 1$, memory is preserved only for small P . However, in the heterogeneous phase, the competition between the different patterns boosts network heterogeneity, and the capacity of the network increases greatly, as it is seen in the new pictures.

Finally, regarding κ_m , we thank the referee for this comment, since it is a typo that should read κ_{∞} , as suggested. We have fixed it.

[1] L G Morelli and M N Kuperman. "Associative memory on a small-world neural network." *The European Physical Journal B-Condensed Matter and Complex Systems* 38:3, 2004.

[2] T Akam and D M Kullmann. "Oscillatory multiplexing of population codes for selective communication in the mammalian brain." *Nat. Revs. Neuroscience* 15:2, (2014).

Figure 3: What do you exactly mean by "obtained from analysis of the control parameters"? How do the transition lines depend on the parameters, e.g. κ_∞ , n , P , a_0 ? Is there an intuitive explanation for why a larger α allows recall even under noise levels above $T > 1$?

In order to obtain figure 3, we have measure m and g along the phase space of the system and both for homogeneous and heterogeneous IC (particularly, in the subspace $0 \leq \alpha \leq 2$, $0 \leq T \leq 2$). We have then identified each phase according the values of these control parameters. For instance, the homogeneous memory phase is characterized by $g=1$ and $m=1$, whereas the homogeneous noisy phase is determined by $g=1$ and $m=0$.

Regarding the influence of the different order parameters, we have found that κ_∞ is not critical whiting realistic values – from $\kappa_\infty = 10$ to approximately 100, although it affects the extend of the bistability region. Results are also robust with respect to the parameter n , taking into account that extremely high values can cause instability of the solution.

We have included a novel section dealing with the effect of P : “Capacity analysis”, starting pg 6, line 406. In summary, results in the main section hold for small P , but the system is driven to the spin glass phase when this number is increased, as expected from other capacity studies in complex networks [1]. In this same section we comment on the effect of a_0 . Our results do not strongly depend on it, but particular distributions of the patterns do affect the dynamics. Following this suggestion of the reviewer, we have included a discussion on the effect of these parameters in the results section starting in line 263, pg 4.

On the other hand, and regarding the second part of this question, the reason why memory is granted by large α even for $T > 1$ is the strong network heterogenization that takes place in this case. As previous studies have shown [2], heterogeneous networks such as scale-free or bimodal ones do not undergo a continuous transition at $T=1$, but it is delayed to $T \rightarrow \infty$ in the thermodynamic limit. Therefore, heterogeneous finite networks can uphold memory up to much higher noise values than their homogeneous counterpart. Given that a high value of α leads very quickly to a very heterogeneous network, it assures the stability of the memory state. We have re-written this discussion in the manuscript (pg 5, starting line 342) in order to make it more clear for the reader.

[1] M Leone et al. "Ferromagnetic ordering in graphs with arbitrary degree distribution." The European Physical Journal B-Condensed Matter and Complex Systems 28:2, 2002.

[2] JJ Torres, MA Munoz, J Marro and PL Garrido. "Influence of topology on the performance of a neural network." Neurocomputing, 58, 229, 2004.

296, 302: How is the clustering coefficient defined and how did you find in the model a decay as k^{-1} ?

As in previous works, we define the clustering coefficient of the nodes as $C_i = \frac{2 * t_i}{k_i * (k_i - 1)}$, where t_i is the number of triangles incident to node i , that is, $t_i = \frac{1}{2} * \sum_{j,h} (a_{ij} * a_{ih} * a_{jh})$. With this we obtain $C(k) = \frac{1}{(N * p(k))} * \sum_i (C_i \delta_{k_i,k})$. This definition has been added in line 462, pg 7.

We have also complemented this section by including a plot of $C(k)$, $k_{nn}(k)$ and $p(k)$ in figure 6, pg 7. We also include a power-law fit of this magnitudes to illustrate the exponents mentioned in the main text (pg 7, starting line 466).

298: Is the nearest neighbour degree function K the same as k_{nn} defined on page 5? If yes, please use the same notation.

Yes, it is the same function. We apologize for this mismatch, which has been addressed in section "Protein interaction networks" (starting pg 7 line 461).

369: "the probabilities of birth and death of synapses ... should be approximately linear". I guess this implies $\alpha \approx \gamma \approx 1$. But on line 183 you argue that only the relation between α and γ matters, which I interpret as $\alpha \approx \gamma$. In this light, the statement in the discussion is not as general as it could be.

We thank the reviewer for this comment. In fact, it is the relation between γ and α what enables the stationary degree distribution to be scale-free, so we have rewritten the statement in the discussion to correctly reflect this result (starting line 588, pg 8).

Suggestions:

Eq 1: The definition of I_i follows many lines later. I think it would help the reader to bring it closer to Eq 1.

We appreciate the suggestion, and we have changed this and moved it to line 133, pg 2 of the new manuscript.

Eq 1 & 3: It could be nice to have the tilde already in Eq 1 and say somewhere that $\sum_i \pi(I_i) = 1$.

We have considered this proposal carefully, but we understand that it would be misleading trying to include $\widetilde{\pi}$ and $\widetilde{\eta}$ in this equation, since their derivation is not straight forward.

On the other hand, we have included the condition that all local probabilities (π , η , $\widetilde{\pi}$ and $\widetilde{\eta}$) are normalized in lines 697 and 704, in pg 9.

Table of page 6: $\widetilde{\eta}(k_i) > \widetilde{\pi}(k_i)$ for $k_i > k$ etc.

We have included this suggestion in the table, which is now in pg 3, starting line 161.

183: The relevant parameter is the ratio α/γ .

We have changed this in the manuscript (this paragraph has been move to the methods: line 778, pg 10).

Paragraph: Synaptic pruning with initial overgrowth: This looks interesting. I think the paper would profit from expanding this part and having more than preliminary simulations.

As they are now, the paragraphs on initial overgrowth and protein interactions feel more like discussion paragraphs than results. Maybe it would be worthwhile to move them to the discussion section.

We agree with the referee that the synaptic pruning with initial overgrowth is an interesting topic. However, we think a comprehensive analysis of this is beyond the scope of this work, in which our main goal is characterizing the coupling between structure and memory. Therefore, we have followed the second suggestion and move this paragraph to the discussion section (starting line 573, pg 8).

On the other hand, we have completed the section on protein interaction networks (“Protein interaction networks”, starting pg 6, line 431) by including a new figure (figure 6, pg 7) showing the clustering coefficient $C(k)$, the nearest neighbors mean degree function, $k_{\text{nn}}(k)$ and the degree distribution for networks in our model. We illustrate how this three magnitudes show a scale-free decay with k for $\alpha \approx$

$\alpha^t c(T)$, and the fit used to obtain its exponents. We understand that with this inclusion the section on protein networks feels more complete.

A somewhat peculiar assumption of the model is that the connection between two neurons i and j is either w_{ij} given by the Hopfield prescription or 0 (when $e_{ij} = 0$), i.e. as soon as a synapse appears between these two neurons its strength is set to a constant value and now growth phase takes place. Maybe you could comment on why this is a reasonable assumption.

We agree with the referee in that this is an unrealistic assumption. We chose to use hebbian synapses since they are an usual way to define memory attractors, and in this work we had interest in analyzing the effect of heterogeneity and the coupling between memory and structure on the dynamics. More realistic scenarios could include time dependent synapses, considering for instance learning [1] of fast noise [2], but this would add more complexity to the model.

We have included a comment on this in the discussion of the manuscript (starting line 537, pg 8), since we think it would an interesting point to consider in future works.

[1] S Song, K D Miller and L F Abbott. "Competitive Hebbian learning through spike-timing-dependent synaptic plasticity." Nature neuroscience 3:9, 2000.

[2] [J M Cortés et al. "Effects of fast presynaptic noise in attractor neural networks." Neural computation. 18:3, 2006.

63: underlying dynamics of I am not an expert on structural plasticity or synaptic pruning, but there seem to be many, also recent, interesting experimental and theoretical studies on this topic that could be worth citing: - Anthony Holtmaat and Karel Svoboda have done interesting experimental studies, e.g.

<http://dx.doi3.org/10.1016/j.neuron.2005.01.00> or
<http://dx.doi.org/10.1038/nrn2721>

On the theory side e.g. <http://dx.doi.org/10.1371/journal.pcbi.1004347>,
<http://dx.doi.org/10.1371/journal.pcbi.1002689> or
<https://arxiv.org/abs/1609.05730>

We thank the referee for pointing out these interesting studies, which we have cited in the introduction. We have found particularly interesting [1],

which we have used to compare our model with experimental data and other model on synaptic pruning.

[2] and [3] also give an interesting approach to the structural changes in the mature brain, which is related to the stationary regime of our model. For instance, as these works point out, there is an ongoing birth and death of synapses in the mature brain, but the statistical properties of link distribution, such as the node's degree distribution, remain constant, as in the stationary state of our model.

We also consider that [4] and [5] propose an interesting insight into spine growth and maturation. Their approach considers spike-timing dependent structural plasticity to explain cooperative synapse formation, and it does not consider the formation of memories.

We have included a paragraph discussing these references in the introduction (starting line 39, pg 1).

[1] S Navlakha, A L Barth, and Z Bar-Joseph. "Decreasing-rate pruning optimizes the construction of efficient and robust distributed networks." *PloS Comput Biol*, 11:e1004347, 2015.

[2] A Holtmaat et al. "Transient and persistent dendritic spines in the neocortex in vivo." *Neuron*, 45:279, 2005.

[3] A Holtmaat, and K Svoboda. "Experience-dependent structural synaptic plasticity in the mammalian brain." *Nature reviews. Neuroscience*, 10:647, 2009.

[4] M Deger, M. Helias, S. Rotter and M. Diesmann. "Spike-timing dependence of structural plasticity explains cooperative synapse formation in the neocortex." *PLoS computational biology* , 8:e1002689, 2012.

[5] M Deger, A. Seeholzer and W. Gerstner. "Multicontact synapses for stable networks: a spike-timing dependent model of dendritic spine plasticity and turnover." *arXiv preprint* , arXiv:1609.05730, 2016.

Reviewers' comments:

Reviewer #1 (Remarks to the Author):

In the revised version of the manuscript authors have addressed all issues raised in the previous review. In my opinion, the manuscript in the current form presents results interesting enough to justify its publication in Nature Communications. However, I suggest a careful language revision, as I could spot quite a few language errors in the new text, e.g.:

- 13-16: There is something wrong with this sentence, probably missing “were” or “have been”,
- 21: “have confirm”,
- 21-22: “actual world neural networks”,
- 185-186: “..., despite previous preliminary studies assumed the opposite [27].”
- 190: “on the opposite”,
- 251: “difficult” used as a verb (?),
- 529: “biestability”,
- 659: “as we proof”.

Reviewer #2 (Remarks to the Author):

With many typos fixed and the provision of additional information, the methods section is now much more accessible.

But the second issue raised in the previous review remains: I don't see how "the concurrence of form and function" provides "an explanation for synaptic pruning". The last paragraph of the discussion sounds interesting. But I don't understand, how the conclusions are obtained. Why would the heterogeneous configuration require too much genetic information? Is initial overgrowth with the extra non-linear, time-dependent factor and $k_0 > k_{\infty}$ the only possibility to reach this configuration from randomly wired neural network? If so, why?

The new section on synaptic pruning recapitulates the results of a previous study by the authors (reference 27), *without* the concurrence of form and function, i.e. in the topological limit. Besides this, the initial overgrowth and subsequent pruning is not an inherent property of the model, but it is just one of many possible behaviours of the model, when the parameters are chosen accordingly, i.e. $k_0 > k_{\infty}$ and including an extra non-linear, time-dependent term in the growth probability.

Currently I am inclined to think that the work of Navlakha et al. (reference 11) provides a more convincing explanation of synaptic pruning.

Also, in the part on protein interaction networks, that gained importance in this version, it is unclear to me, how the concurrence of form and function is necessary to fit the experimental data. Is the fit also possible in the topological limit?

In conclusion, it is unclear to me, what the main message of this paper is. The phase diagram of the coupled system is a small, but interesting contribution. But unfortunately it is still unclear to me what this result implies for synaptic pruning and protein interaction networks.

Minor points:

- * What is the definition of "patch" in the capacity analysis?
- * The tick labels of the conceptual age axis in figure 2 are copied wrongly from the original publication!
- * The expectation over time in the definition of m is mentioned only in the response to my previous review, not in the method section.
- * There are still quite a few typos.

We very much appreciated the reports of both reviewers, which have helped us to further improve the manuscript. We are particularly thankful to reviewer 1 for his/her positive assessment on our work. We have carefully revised the manuscript so as to address possible language errors and typos to the best of our capabilities. We also thank reviewer 2 for his/her thoughtful comments, which have led us to undertake further research, improving upon our initial results significantly. In this version of the manuscript we have highlighted in blue all the relevant new or re-structured text. We address the full review in detail below, where our comments have been written in blue.

Reviewer #2:

With many typos fixed and the provision of additional information, the methods section is now much more accessible.

But the second issue raised in the previous review remains: I don't see how "the concurrence of form and function" provides "an explanation for synaptic pruning". The last paragraph of the discussion sounds interesting. But I don't understand, how the conclusions are obtained. Why would the heterogeneous configuration require too much genetic information? Is initial overgrowth with the extra non-linear, time-dependent factor and $k_0 > k_{\infty}$ the only possibility to reach this configuration from randomly wired neural network? If so, why?

We agree that these points were not sufficiently well explained in the initial manuscript, and we have endeavored to improve the text in this regard. We have also obtained some significant new results which, as we go on to describe, bolster the view that synaptic pruning is indeed related to the mechanism we propose.

When we stated that building an initial neural network with particular topological properties from birth would require too much genetic information, we were referring to the fact that there are too many neurons and synapses for the exact properties of each to be specified genetically, and that a process which began with a more or less random network and gradually tuned it via synaptic pruning would be much more efficient. This is a view shared by many neuroscientists (such as Kolb and Gibb, 2011; our new Ref [56]), and would seem, at least implicitly, to underlie other work on synaptic pruning (like Navlakha *et al*, Ref [21]). However, we agree that it would presumably require much less genetic information to specify a broad degree distribution than, say, a given degree sequence. Thus, while a mechanism such as we propose might be sufficient to fashion a scale-free network from a fully random one, it does not appear to be necessary. However, we have now included a new result in the section 'Capacity analysis', plus some interpretation thereof in 'Discussion', which puts this in a new perspective. We show that the networks generated with the coupled model have much better memory performance, *for the particular memory patterns that were learned at the outset*, than the scale-free networks generated with the topological (i.e. uncoupled) version of the model. This is a property which cannot be genetically encoded, since the specific memory patterns which an infant neural network will need to learn are not known before it develops.

Regarding the second part of the question, we would like to note that we do not claim that the inclusion of the extra (growth) term is the only possibility to reach a heterogeneous configuration. However, this initial overgrowth does appear in most of the experimental data concerning synaptic pruning reported in the literature [1,2]. Similarly, $k_0 > k_{\infty}$ in the experimental data we have fitted, so this is not an assumption of the model, but a natural result of fitting the data. In the discussion section we also argued why these experimentally motivated considerations could serve as a simple way to maintain memory during infancy and synaptic pruning. This has been more

clearly expressed in the new version of the manuscript in the “Synaptic pruning” and “Discussion” sections.

[1] M S Keshavan, S Anderson, and J W Pettergrew. “Is schizophrenia due to excessive synaptic pruning in the prefrontal cortex? the Feinberg hypothesis revisited.” *Journal of Psychiatric Research*, 28:239, 1994.

[2] Kolb, B., Mychasiuk, R., Muhammad, A., Li, Y., Frost, D. O. and Gibb, R. “Experience and the developing prefrontal cortex.” *Proceedings of the National Academy of Sciences*, 109, 2012.

The new section on synaptic pruning recapitulates the results of a previous study by the authors (reference 27), *without* the concurrence of form and function, i.e. in the topological limit. Besides this, the initial overgrowth and subsequent pruning is not an inherent property of the model, but it is just one of many possible behaviours of the model, when the parameters are chosen accordingly, i.e. $k_0 > k_{\infty}$ and including an extra non-linear, time-dependent term in the growth probability.

We agree with the Reviewer’s positive assessment that our model provides a simple and general theoretical framework in the form of a master equation. This could allow us to explore the emergent properties induced in a developing network when different microscopic mechanisms are included – for instance, one could study different probabilities to drive pruning dynamics. Under our view this is a clear indication of its possible value, relevance and impact for the scientific community.

We also note that in the present work we have fitted two extremely different experimental data-sets: they are reported from different authors, and have been recorded in different animals and obtained using different experimental techniques, and only one of them had already been analyzed [27].

Moreover, in both cases the parameters used to reproduce the initial overgrowth and the pruning have been chosen according to the experimental data, in which $k_0 > k_{\infty}$, so this is not an arbitrary decision.

Currently I am inclined to think that the work of Navlakha et al. (reference 11) provides a more convincing explanation of synaptic pruning.

We do not view the work of Navlakha *et al* as in any way incompatible with ours, but we believe the mechanism we describe adds an important ingredient to the explanation. Navlakha and co-authors showed that, beginning with a dense but random structure, network topology can be optimized for efficiency and robustness by removing specific edges according to a rule. Our mechanism does something similar, except that the optimization is for memory performance. However, Navlakha’s rule requires, in the words of the authors, a ‘feedback to the circuit that “rewards” every edge active along a source-to-target response’ - in other words, a form of back-propagation. They leave open what biological mechanism might provide this non-local feedback. By coupling a neural network model to an evolving network structure, we have shown that such back-propagation is not required: synapses need only respond to their local currents, as they have been shown to do in reality [1,2]. (Although it falls outside the remit of this work, it would perhaps be interesting to study which local rules of this kind lead, macroscopically, to the kind of rule Navlakha *et al* considered.)

We believe that our work adds a significant element to our understanding of how synaptic pruning can act as an optimization mechanism for neural network performance. However, upon reflection we have decided that it might be an oversell to call this an “explanation”, as though there were nothing left to explain, so in the revised manuscript we speak merely of the role this mechanism might play in synaptic pruning.

- [1] K S Lee, F Schottler, M Oliver and G Lynch. "Brief bursts of high-frequency stimulation produce two types of structural change in rat hippocampus." *Journal of Neurophysiology*, 44:247, 1980.
- [2] M De Roo, P Klauser, P Mendez, L Poglia and D Muller. "Activity-dependent PSD formation and stabilization of newly formed spines in hippocampal slice cultures." *Cerebral Cortex*, 18:151, 2008.

Also, in the part on protein interaction networks, that gained importance in this version, it is unclear to me, how the concurrence of form and function is necessary to fit the experimental data. Is the fit also possible in the topological limit?

Although the Reviewer is right and the protein data we show could be fitted by the topological model too, our results show that the physiological model reported in our work expands the region in which scale-free networks are obtained (in the topological limit this is only for $\alpha = 1$). Therefore, our model is more robust with respect to small changes of parameters such as α and the noise level σ , a fact that could yield advantages for noisy interacting systems such as protein interaction networks, where the influence of noise has been reported too.

Moreover, it has recently been revealed in multiple studies that in protein interaction networks there are specific protein interaction patterns that can be studied, and that would allow us to identify important biological substructures in the network [1,2], or to identify frequently occurring interaction patterns in its functional space [3,4]. This information could be introduced in the model proposed here to determine the relevance of such patterns and to study the complex interplay between their functional role and the underlying network structure.

In the revised version of our manuscript, we have included a new paragraph discussing this important issue on section "Results: Protein interaction networks".

- [1] Wei Xiong and Luyu Xie and Shuigeng Zhou and Jihong Guan, "Active learning for protein function prediction in protein-protein interaction networks", *Neurocomputing*, 145 (2014).
- [2] Jun Ren, Jianxin Wang, "Identifying protein complexes based on density and modularity in protein-protein interaction network" *BMC Systems Biology*, 7 (2013).
- [3] Mehmet E Turanalp and Tolga Can, "Discovering functional interaction patterns in protein-protein interaction networks" *BMC Bioinformatics* 9:276 (2008)
- [4] Guangming Liu, Huixin Wang, Hongwei Chu, Jian Yu & Xuezhong Zhou, "Functional diversity of topological modules in human protein-protein interaction networks" *Scientific Reports* 7, 16199 (2017).

In conclusion, it is unclear to me, what the main message of this paper is. The phase diagram of the coupled system is a small, but interesting contribution. But unfortunately it is still unclear to me what this result implies for synaptic pruning and protein interaction networks.

We agree with the Reviewer that in the previous version of the manuscript the main message of our work was not sufficiently clear. We hope that, after the many changes we have made, the Reviewer would now form a different opinion. These changes include the new results regarding optimization for specific patterns, in the section 'Capacity analysis', and many parts of the text which have been rewritten. The latter are shown in blue font in the manuscript, and include changes to most of the sections (including the title, abstract and discussion).

Minor points:

* What is the definition of "patch" in the capacity analysis?

A "patch" referred to a small set of active neurons. Since this could be confusing, we have changed the name to be just "small set of active neurons", and avoid any possible confusion.

* The tick labels of the conceptual age axis in figure 2 are copied wrongly from the original publication!

We are sorry for this mistake, which has been addressed.

* The expectation over time in the definition of m is mentioned only in the response to my previous review, not in the method section.

This issue has now been fixed. We have adapted the notation and defined \bar{m} as the expectation over time of $m(t)$ for $t \gg 1$; and similarly with the topological variables $g(t)$ and $r(t)$.

* There are still quite a few typos.

We thank the referee for pointing this out, and we have worked on the readability and grammar of the text.

REVIEWERS' COMMENTS:

Reviewer #2 (Remarks to the Author):

I thank the authors for carefully addressing all my concerns. I think the manuscript is now ready for publication.

REVIEWERS' COMMENTS:

Reviewer #2 (Remarks to the Author):

I thank the authors for carefully addressing all my concerns. I think the manuscript is now ready for publication.

AUTHORS RESPONSE:

We very much appreciated the throughout reviews of the referees during the revision of the manuscript, which have significantly contributed to improving the quality of the publication.